


# 1 Atmospheric H₂ observations from the NOAA Global
# 2 Cooperative Air Sampling Network

Gabrielle Pétron[1,2], Andrew M. Crotwell[1,2], John Mund[1,2], Molly Crotwell[1,2], Thomas Mefford[1,2],
Kirk Thoning[2], Bradley Hall[2], Duane Kitzis[1,2], Monica Madronich[1,2], Eric Moglia[1,2], Don Neff[1,2],
Sonja Wolter[1,2], Armin Jordan[3], Paul Krummel[4], Ray Langenfelds[4], John Patterson[5]
*Correspondence to*: Gabrielle Pétron (gabrielle.petron@noaa.gov)
1. Cooperative Institute for Research in Environmental Sciences, CU Boulder, USA
2. NOAA Global Chemical Laboratory, Boulder, USA
3. Max-Planck-Institute for Biogeochemistry (MPI-BGC), Jena, Germany
4. Commonwealth Scientific and Industrial Research Organisation - Environment, Aspendale, Australia
5. Department of Earth System Science, University of California, Irvine, USA
**Abstract.** The NOAA Global Monitoring Laboratory measures atmospheric hydrogen ($H_2$) in
grab-samples collected weekly as flask pairs at over 50 sites in the Global Cooperative Air Sampling
Network. These NOAA $H_2$ measurements from 2009 to 2021 are publicly available. Measurements
representative of background air sampling show higher $H_2$ in recent years at all latitudes. The marine
boundary layer (MBL) global mean $H_2$ was 20.2 ±0.2 ppb higher in 2021 compared to 2010. A 10 ppb or
more increase over the 2010-2021 average annual cycle was detected in 2016 for MBL zonal means in the
tropics and in the Southern Hemisphere. Carbon monoxide measurements in the same air samples suggest
large biomass burning events in different regions likely contributed to the observed interannual variability
at different latitudes. A major focus in recent years involved the adoption of the World Meteorological
Organization Global Atmospheric Watch (WMO GAW) $H_2$ mole fraction X2009 calibration scale,
developed and maintained by the Max-Planck Institute for Biogeochemistry (MPI-BGC), Jena, Germany.
GML maintains eight $H_2$ primary calibration standards to propagate the MPI scale. These are gravimetric
hydrogen-in-air  mixtures in electropolished stainless steel cylinders (Essex Industries, st. Louis, MO)
which are stable for $H_2$. These mixtures were calibrated at the MPI-BGC, the WMO Central Calibration
Laboratory (CCL) for $H_2$, in late 2020 and span the range 250-700 ppb. We have used the CCL
assignments to propagate the MPI X2009 $H_2$ calibration scale to NOAA air measurements performed
using Gas Chromatography-Helium Pulse Discharge Detector instruments since 2009. To propagate the
scale, NOAA uses a hierarchy of secondary and tertiary standards, which are high pressure tanks with
whole air mixtures calibrated against the primary and secondary standards respectively. NOAA secondary
and tertiary standards are stored in aluminum cylinders, which have a tendency to grow $H_2$ over time. We
fit the calibration histories of these standards with 0-2nd order polynomial functions of time and use the
time-dependent mole fraction assignments on the MPI X2009 to reprocess all tank air and flask air
measurement records. The robustness of the scale propagation over multiple years is evaluated with the
regular analysis of target air cylinders and with long-term same air measurement comparison efforts with
WMO GAW partner laboratories. Long-term calibrated, globally distributed and freely accessible
measurements of $H_2$ and other gasses and isotopes continue to be essential to track and interpret regional
and global changes in the atmosphere composition. The adoption of the MPI X2009 $H_2$ calibration scale
and subsequent reprocessing of NOAA atmospheric data constitute a significant improvement in the
NOAA $H_2$ measurement records.



## 1 Introduction

High quality and sustained observations are essential to track and study changes in atmospheric trace gas distributions. Ambient air measurement programs for trace gasses provide objective data to track air pollution levels [Oltmans and Levy, 1994; Thomson et al., 2004; Tørseth et al., 2012; Schultz et al., 2015; Cooper et al., 2020; WMO, 2022], to study how a mix of sources (and sinks) impact the air composition [Ciais et al., 1995; Langenfelds et al., 2002; Brito et al., 2015] and to constrain and evaluate fluxes and their trends [von Schneidemesser et al., 2010; Simpson et al., 2012; Propper et al., 2015; Montzka et al., 2018; Friedlingstein, 2022; Heiskanen et al., 2022; Storm et al., 2023] at scales of interest.

$H_2$ is a trace gas in the Earth's atmosphere and its abundance can indirectly impact climate and air quality. The analysis of $H_2$ measurements in firn air collected in Antarctica reveal that $H_2$ levels in the high southern hemisphere grew by some 70% (330 to 550 ppb) over the 20th century [Patterson et al., 2021; 2023]. Greenland firn air covers less depth and time but results are consistent with a 30% increase in high northern hemisphere $H_2$ from 1950 to the late 1980s [Patterson et al., 2023]. Growing emissions related to fossil fuel burning most likely were behind this rise in $H_2$ [Patterson et al., 2021]. Results also show that $H_2$ in both polar regions leveled off after the 1990s [Patterson et al., 2021, 2023].

$H_2$ has been viewed as a potential low or zero carbon energy carrier for close to five decades [Yap and McLellan, 2023]. Since 2020 there has been renewed interest in the hydrogen economy [Yap and McLellan, 2023] spurred by a rise in announcements of public and private projects to produce low carbon $H_2$, also referred to as "blue" $H_2$ produced from natural gas with carbon capture, utilization and storage, or "green" $H_2$ produced using renewable energy [Hydrogen Council and McKinsey & Company, 2023]. In 2021, $H_2$ global demand was over 94 million tonnes or 2.5 % of global final energy consumption [IEA, 2022]. This demand was almost entirely driven by refineries and a few industries (ammonia, methanol and steel) and $H_2$ production almost entirely relied on fossil fuels with unabated emissions ("gray $H_2$", [IEA, 2022]). As of December 2023, over 1,400 announced projects globally (worth US$ 570 billion) are anticipated to increase the global $H_2$ production capacity by 45 million tonnes through 2030 [Hydrogen Council and McKinsey & Company, 2023].

Studies of the potential short-term and long-term climate impacts of increased $H_2$ production and use have called for more research to better understand the current and future $H_2$ supply chain and end-use emissions of $H_2$ and GHGs [Ocko and Hamburg, 2022; Longden et al., 2022; de Kleijne et al., 2022; Bertagni et al., 2022]. Global, high quality and sustained atmospheric measurements of $H_2$ can provide independent information to document its distribution and study its sources and sinks and how they may change.

The National Oceanic and Atmospheric Administration (NOAA) Global Cooperative Air Sampling Network comprises over 50 surface and mostly remote sites (https://gml.noaa.gov/ccgg/flask.html). At each site and on a weekly basis, local partners collect air in two 2.5-L glass flasks, and then return the flasks to the NOAA Global Monitoring Laboratory (GML) in Boulder, Colorado, USA, for measurements of major long-lived greenhouse gasses, carbon dioxide ($CO_2$), methane ($CH_4$), nitrous oxide ($N_2O$), sulfur hexafluoride ($SF_6$), as well as carbon monoxide (CO) and hydrogen ($H_2$) [Conway et al., 1994; Novelli et al., 1999; Dlugokencky et al., 2009]. The network is a contributor to the World Meteorological





Organization (WMO) Global Atmospheric Watch (GAW) Programme, which promotes and coordinates
international scientific efforts and free access to long-term atmospheric observations [WMO, 2022].
CO and $H_2$ are important trace gasses that share sources with $CO_2$ and $CH_4$ (fossil fuel burning, biofuel
burning and wildfires). Reaction with hydroxyl radicals (OH) is the main sink for $CH_4$ and CO and an
important sink for $H_2$. Both $H_2$ and CO are also produced during the chemical oxidation of $CH_4$ and
nonmethane hydrocarbons. Soil uptake by bacteria accounts for 75% of the total $H_2$ sink. $H_2$ and CO have
much shorter atmospheric lifetimes than $CO_2$ and $CH_4$: 2-3 months for CO and close to 2 years for $H_2$.
The $H_2$ global mean atmospheric lifetime is largely driven by the soil sink strength. The $H_2$ lifetime
related to the oxidation by OH is estimated to be 8-9 years [Price et al., 2007; Warwick et al., 2022].
The "Geophysical Monitoring for Climatic Change" was the original program established by NOAA to
gather and analyze observations of the background atmosphere composition. GMCC started measuring
$CO_2$ in background air samples in 1968 [Komhyr et al., 1985]. $CH_4$ was added in 1983 [Steele et al.,
1987].  In the late 1980s, GMCC and its successor the Climate Monitoring and Diagnostics Laboratory
expanded operations to measure CO in the global network air samples to add a constraint for the study of
combustion sources and the global carbon budget. The analytical instrument selected consisted of a gas
chromatograph (GC) and a reduction gas analyzer (RGA, from Trace Analytical Inc., California) that
could measure both CO and $H_2$.
Novelli et al. [1992, 1991] first reported on testing the sampling approach (flask type, stopcock fitting,
wet/dry air, untaped versus taped glass flasks to minimize direct sunlight exposure) and GC-RGA
instrumentation for CO. Around that time, other laboratories had also adopted the GC-RGA measurement
technique for CO and $H_2$ measurements in discrete air samples or in situ. Khalil and Rasmussen [1989,
1990] reported on $H_2$ measurements of whole air samples collected weekly in triplicate electropolished
stainless steel flasks between October 1985 and April 1989 at the four NOAA atmospheric baseline
observatories (Point Barrow, Mauna Loa, Samoa, South Pole), Cape Meares, Oregon, Cape Kumukahi,
Hawaii and at the Cape Grim Observatory, Tasmania. These measurements showed that, contrary to $CO_2$,
$CH_4$, $N_2O$ and CO, background air $H_2$ levels were higher in the Southern Hemisphere (SH) than in the
Northern Hemisphere (NH). 1985-1987 monthly mean observed $H_2$ ranged between 500-520 ppb at the
South Pole and between 455 and 520 ppb at Point Barrow. $H_2$ exhibited a strong seasonal cycle at
extratropical latitudes especially in the NH and the seasonal cycles in both hemispheres were offset by 1-2
months only.
In 1995, $H_2$ standards were prepared gravimetrically in Scott Marrin Inc. cylinders and five of them
(spanning 485-603 ppb) were used to define the NOAA $H_2$ X1996 calibration scale. Working standards
used between 1988 and 1996 were reassigned $H_2$ mole fractions and flask air measurements were
reprocessed to be on the X1996 scale. Novelli et al. [1999] described the early NOAA $H_2$ measurements
and reported $H_2$ time series starting in the late 1980s or early 1990s (depending on the site) for 50 sites in
the NOAA Cooperative Global Surface Air Sampling Network.
Simmonds et al. [2000] reported in-situ high-frequency GC-RGA3 measurements of $H_2$ at the Mace Head
baseline atmospheric monitoring station on the Atlantic coast of Ireland for the 1994-1998 period. They
found that the background air at Mace Head had lower monthly mean $H_2$ (470-520 ppb) than background



air masses measured at the Cape Grim observatory (510-530 ppb) from July to April. Some of the 40 min
$H_2$ observations showed 10s-200 ppb short-term $H_2$ enhancements above baseline levels. The authors
derived an estimate of European emissions with an inverse model of enhanced $H_2$ in air masses impacted
by upwind sources of pollution. They also observed that nighttime measurements in low wind conditions
reflected local depletion of $H_2$. The authors derived variable mean deposition velocities and found that the
$H_2$ soil sink was likely a process that occurred year-round in the area.
After 1996 and until 2008, the NOAA $H_2$ measurement program used successive working standards that
were assigned based on GC-RGA measurements against the previous standards. With hindsight, the
NOAA X1996 calibration scale transfer and the early NOAA $H_2$ measurements had several limitations
which are briefly described below and in more detail in the Supplementary Information section S1.
By the late 1990s, same air or collocated air sample measurement comparison between NOAA and the
Commonwealth Scientific and Industrial Research Organisation (CSIRO) for the Cape Grim Observatory
and Alert, Canada, flask air analyses showed an increasing bias for $H_2$ between the two laboratories
[Masarie et al., 2001; Francey et al., 2003]. Further laboratory tests by several WMO/GAW measurement
laboratories revealed the RGA detector response was non linear and required frequent calibration.
Additionally measurement laboratories found that the $H_2$ mole fraction for air standards, especially those
stored in high pressure aluminum cylinders, could drift at rates of a few parts per billion (ppb) to tens of
ppbs per year [Novelli et al., 1999; Masarie et al., 2001; Jordan and Steinberg, 2011].
To address these compounding issues, in 2008 GML tested a new analytical instrument: a gas
chromatograph with a pulse discharge helium ionization detector (GC-HePDD) [Wentworth et al., 1994].
The technique showed very good performance with a stable and linear response over the 0-2000 ppb
range and it was adopted for the calibration scale propagation and flask air analysis in 2009 [Novelli et
al., 2009]. Around that time NOAA also began testing electropolished stainless steel cylinders (Essex
Industries, St. Louis, MO) filled with dry air for stability.
In 2007-2008, GML prepared six new gravimetric air mixtures in electropolished stainless steel cylinders
spanning 250-600 ppb $H_2$. At that time, the new gravimetric mixtures differed by about +20 ppb
compared to two $H_2$ secondary standards values assigned on the NOAA $H_2$ X1996 scale. For the next
decade, GML kept using the X1996 calibration scale while also conducting routine measurements of the
$H_2$ secondary standards against the 2007/2008 gravimetric mixtures.
The GC-HePDD $H_2$ measurements on the NOAA $H_2$ X1996 remained biased compared to GAW partner
measurements and the NOAA $H_2$ data from the global network flasks were not released publicly. SI
sections S1-3 provide background information on issues impacting the 1988-2008 NOAA $H_2$
measurements on RGAs, and related information from the CSIRO and Max-Planck Institute for
Biogeochemistry (MPI-BGC) $H_2$ measurement programs. The best NOAA $H_2$ measurement records date
back to 2009/2010 and are the main focus of this paper.
In Fall 2020, GML initiated an effort to 1) adopt the MPI X2009 $H_2$ calibration scale [Jordan and
Steinberg, 2011] for future measurements and 2) convert GML $H_2$ measurements made on GC-HePDD
instruments (beginning in late 2009) to that scale. This paper describes the MPI X2009 $H_2$ calibration





scale propagation within GML and the revised measurements from the NOAA Global Cooperative Air
Sampling Network flask air samples analyzed since late 2009. We show very good agreement for the
reprocessed NOAA $H_2$ data for different WMO/GAW measurement comparison efforts. The revised
NOAA GML flask air $H_2$ dry air mole fraction measurement records for 70 surface sites from 2009-2021
are publicly available [Pétron et al., 2023a]. This new dataset complements other WMO/GAW $H_2$
measurement datasets and provides reliable observational constraints for the study of atmospheric $H_2$
global distribution and budget since 2009. Future NOAA $H_2$ dataset updates will be released as we use
continued calibration results to reliably track the drift in working standards and revise their assignments.

## 2 Adoption of the MPI X2009 $H_2$ calibration scale

In this section, we introduce the GML $H_2$ calibration standard hierarchy. First we introduce the GML $H_2$
primary standards. Then we describe the GML tank air $H_2$ calibration system and the scale transfer from
the primary standards to secondary and tertiary standards (2009-April 2019) or from the primary
standards to working standards (after April 2019). The tertiary standards and working standards are used
to calibrate the $H_2$ instrument response on the flask air analysis systems and value assign discrete air
measurements. An important quality assurance procedure within GML is the routine measurement of
dedicated quality control cylinders (referred to as "Target" tanks) to track instrument performance.
Results are discussed in relation to the uncertainty of the flask air analysis systems and consistency of the
MPI X2009 $H_2$ scale implementation.

### 2.1 GML $H_2$ primary standards

In 2007-2008, six air mixtures spanning a range of $H_2$ dry air mole fractions were prepared
gravimetrically in electropolished stainless steel 34L cylinders ([Novelli et al., 2009], and Table 1). The
highest $H_2$ mole fraction tank developed a leak and was lost. The remaining set of five standards covered
the range 250 ppb to 600 ppb for $H_2$. Three standards in electropolished stainless steel cylinders were
added in 2019 to extend the upper limit of the calibration range to 700 ppb $H_2$ and evaluate the stability of
the initial set over the intervening years. These eight standards have been designated by GML as our
highest level $H_2$ standards. We refer to them as the NOAA $H_2$ primary standards throughout this document
even though they are not defining the scale.
The eight primary standards were analyzed by the WMO Central Calibration Laboratory (CCL) for $H_2$
hosted by the MPI-BGC in Jena, Germany, on their GC-PDD system in November 2020. The results
listed in Table 1 are reported on the MPI X2009 $H_2$ calibration scale [Jordan and Steinberg, 2011]. The
CCL uncertainty estimates listed in Table 1 refer to the standard deviation of the 25-32 discreet $H_2$
measurements made for each standard. Until they are recalibrated by the CCL, we add an 0.5 ppb 1-sigma
uncertainty to these assignments. This is the currently reported CCL reproducibility for their GC-HePDD
$H_2$ measurements. It accounts for potential longer term uncertainty in calibration results that would not be
evident in the standard deviations of measurements made close in time.

### 2.2 MPI X2009 $H_2$ calibration scale transfer


GML has separate, dedicated analytical systems for scale propagation and flask air analyses. Novelli et al. [2009] describe the GC-HePDD instruments and the operating parameters in detail. GML has used three GC-HePDD instruments so far. Each is identified by a unique internal instrument id: H9 (Agilent 6890 GC, serial number US10326037) for tank calibrations and H8 (Agilent 6890 GC, serial number US10326011) and H11 (Agilent 7890 GC, serial number US10834030) for flask analyses. The GC-HePDD instruments' responses are linear (within 0.3%) up to 2000 ppb. They are configured for ppb level sensitivity and calibrated over the 200-700 ppb range, which is optimal for global and regional background air analysis.

The GML $H_2$ primary standards are used to periodically calibrate the H9 instrument response for the analysis and value assignment of lower level standards. GML used secondary standards from 2009 through April 2019 in the calibration hierarchy but has since removed this level to reduce the number of standards which can potentially drift (see discussion of drifting cylinders later in this document). The stability and longevity of the primary standards are critical to ensure the consistency of the GML $H_2$ measurements over long periods of time as required for trend analysis.

The $H_2$ secondary and tertiary (or working standards) used in GML are whole air mixtures in high pressure aluminum cylinders (Luxfer USA). Most were filled at the GML standard air preparation facility at the Niwot Ridge mountain research station using a Rix Industries (Benicia, CA) SA6 oil-free compressor [Kitzis, 2017]. Two additional tertiary standards (CB11551 and CC305198) were purchased from Scott Marrin. All GML tank air mixtures have a unique combination of an alphanumeric cylinder ID and a fill code letter (A-Z) tied to a fill date.

Aluminum tanks are known to be unstable for storing $H_2$ in air standards [Jordan and Steinberg, 2011]. Therefore regular analyses of working standards on the tank calibrations system are critical for quantifying drift to allow a time dependent value assignment on the X2009 $H_2$ calibration scale.

The calibration history for a secondary, tertiary or working standard only uses retained (valid) calibration event results on H9. GML uses python software developed in house to write instrument output files and to calculate a calibration event result. Another piece of python code is used to analyze tank air calibration histories and evaluate if the $H_2$ mole fraction in the tank is stable or if it changes over time. For many GML $H_2$ calibration standards and target air tanks, a linear or quadratic function is the best fit through their calibration history. When this happens, the function coefficients define the tank time-dependent assignment. All mole fraction assignments for standards used to propagate a calibration scale are stored in a database table that can be accessed by the data processing software. As new calibration results are available, the drift correction and assignment for a particular tank ID and fill code is revised as needed and the affected data is reprocessed.

### 2.2.1 Tank calibration system 2009-2019 configuration

From 2007 through mid-April 2019, the $H_2$ tank air calibration on the H9 instrument was conducted using a single standard gas (primary or secondary standard) to calibrate the "unknown" (secondary or tertiary) standards. Each calibration event consisted of alternating injections of the reference gas and the "unknown" with typically seven or more unknown air injections. The first aliquot in a multi injection



measurement sequence on H9 is often slightly biased (due to subtle timing differences with the regulator
flush cycle) and is not used. The ratio of the $H_2$ peak height for each valid "unknown" air injection and
the mean peak height of two bracketing reference gas injections (or sometimes only one preceding or
following reference gas injection) is multiplied by the reference/standard gas known $H_2$ mole fraction to
calculate the "unknown" air injection mole fraction. Results for a tank air calibration event are defined by
the mean and the standard deviation of the calculated $H_2$ mole fractions for five or more retained
unknown air injections. Typically, the standard deviation for a tank air calibration event on H9 is less than
1 ppb.
Prior to the 2023 GML $H_2$ data reprocessing, GML used peak area for the GC-HePDD as described in
Novelli et al. [1999]. However, we saw that for some Helium carrier gas tanks (Airgas Ultra High Purity,
(99.999% purity), the $H_2$ chromatogram peak had a tail or a noisy baseline. Since the $H_2$ peak height was
less affected, we use peak height ratios for all GC-HePDD measurements. In 2023, GML switched to
Matheson Research Grade Helium carrier gas for the GC-HePDDs (99.9999% purity).
Two secondary standards with background ambient air level $H_2$ were in service on H9 to calibrate tertiary
standards: CC119811 (2008/02 to 2013/06) and CA03233 (2013/06 to 2018/11). These two standards
were calibrated periodically on H9 against individual members of the primary standard suite. Most
calibration episodes consisted of one to 6 calibration sequences over 1-3 successive days, each against
one of the primary standards. For CC119811, 1-point calibration sequence results in 2008, 2009, 2010 and
2013 against one of the two lowest primary standards (SX-3558 and SX-3543) show a 3 to 5 ppb positive
bias which suggests a small non zero intercept in the instrument response during those times. This
primary standard dependent bias is not apparent for CA03233 results between 2014 and 2016. Results
against SX-3558 were not used for value assigning either secondary standards and results against
SX-3543 were not used for CC119811.
The calibration results for the two $H_2$ secondary standards are plotted in Figure 1 and final assignments
are listed in Table 2. CA03233 was stable for $H_2$ over its time of use and has an assignment of 502.8 ppb
$H_2$. $H_2$ in CC119811 exhibited a small linear drift and its value assignment is time dependent with a
growth rate of 2 ppb/yr. Between 2009 and 2019, the two secondary standards were used on H9 to
calibrate seventeen $H_2$ tertiary standards used in the NOAA discrete air sample analysis laboratory.

**2.2.2 Tank air calibration system 2019-present configuration**

Beginning in April 2019, GML transitioned H9 to use a multi-point calibration strategy to better define
the instrument response. The eight $H_2$ primary standards are measured relative to a reference air tank
(CC49559, filled at ambient Niwot Ridge air) to calibrate the instrument response. A multi-standard
response calibration episode for H9 involves the alternating injections from the reference air tank and
each primary standard. Each standard is injected 8 times alternating with reference air aliquots. The entire
response calibration sequence takes close to 15 hours. GML has performed an H9 instrument response
calibration followed by tank calibrations a few times a year over a 10-14 day period each time.
The H9 instrument response function is calculated as the best linear fit to the primary standards' mean
normalized chromatogram peak heights and their CCL $H_2$ mole fraction assignments. H9 calibration





curves are assumed to be valid for several weeks during which time other air cylinders are analyzed
relative to the same reference tank.
Between April 2019 and December 2022, H9 instrument response was calibrated nine times. Figure 2a
shows the deviations of the H9 linear response functions from the line defined by computing the mean
value for the intercept and slope of the 2019-2022 response functions. The instrument response has
remained stable within +/- 1 ppb over this time period over the range 200-700 ppb. Figure 2b shows the
residuals to the best linear fit for each instrument response calibration episode. We note that the H9
instrument response has been quite stable over the 200-700 ppb range but that the linear fit does not go
through the origin. The residuals to the linear fit over this time period are all within the -0.6 ppb to 0.5
ppb range. The linear fit y-intercept ranges between 3.9 and 5.5 ppb (not shown). Prior to 2019, we
assumed a zero intercept for the H9 one point calibration. If we assume a y-intercept around 5 ppb was
more likely, it is possible the pre-2019 H9 measurements (with 1 point calibration) were biased by ~1% of
the difference between the tank air and the standard $H_2$ mole fractions. We do not correct for this potential
bias at this time.
A tank air measurement sequence typically consists of 7 injections, each bracketed by reference air
injections. The peak heights for the first injections of reference air and tank air on H9 can have a small
low bias and are not used. The normalized peak heights for the valid tank air injections are converted to
$H_2$ mole fractions using the most recent H9 instrument response calibration episode. The average and
standard deviation of the retained injection $H_2$ mole fractions are stored in a database table.
**2.2.3 $H_2$ standards and calibration approach for the flask air analysis system**
The NOAA Global Cooperative Air Sampling Network dates back to 1967. In recent years, it has
included over 50 surface sites distributed around the world
(https://gml.noaa.gov/ccgg/behind_the_scenes/network.html). Partners at each site collect air sample pairs
in two 2.5L glass flasks filled simultaneously once a week and return the samples to NOAA GML in
Boulder. $H_2$ in those flask air samples is measured in addition to long-lived GHGs ($CO_2$, $CH_4$, $N_2O$, $SF_6$)
and CO by the Measurement of Atmospheric Gases that Influence Climate Change (MAGICC) system in
the NOAA GML Boulder laboratory. Until mid 2019, GML operated two nearly-identical automated flask
air analytical systems: MAGICC-1 (1997-2019) and MAGICC-2 (2003-2014). Since mid-2019, GML has
used a new MAGICC-3 system. This new system improved analytical techniques for $CO_2$, $CH_4$, $N_2O$, and
CO but continues to use the same GC-HePDD instruments from the older systems.
Two GC-HePDD instruments have been used for hydrogen analysis on the three flask air analysis systems
since 2009: H8 (MAGICC-2: 2009-2014 and MAGICC-3: August 2019-September 2020) and H11
(MAGICC-1: 2010-July 2019 and MAGICC-3: September 2020-present).
On MAGICC-1 and MAGICC-2, the He-PDD instrument response was calibrated using a single tertiary
standard (measured before and after each sample aliquot), similar to the original approach used on H9.
Typically, the $H_2$ tertiary standards used during that time lasted less than a year and most displayed $H_2$
growth over time. Figure 3 shows the calibration histories for H8 and H11 tertiary standards and their
start/deployment dates. Table 2 provides a list of the standard cylinder IDs and fill codes and information





for their mole fraction assignments: t0 date, the best polynomial function fit coefficients relative to time t0
(ci, i=0,2) and an estimated 1-sigma uncertainty. The uncertainty is empirically derived and based on the
standard calibration history and the standard deviation of the residuals to the best fit (the assignment). The
python code used to calculate the tertiary standard assignment uses a 0.5 ppb 1-sigma reproducibility
uncertainty which is added in quadrature to the measurement episode standard deviation to account for
longer term uncertainties not evident in the standard deviation of the n-aliquots. We do not formally
include an uncertainty for the secondary standard assignment. The H9 reproducibility term is based on the
mean of the standard deviation of residuals to the fit for the calibration histories of secondary standards
and target tanks over the period 2008-2022 (see section 2.3.1).
The number of tertiary standards used successively on the older systems introduces time dependent issues
due to the variable rate of $H_2$ drift in aluminum tanks and the frequency of the tank calibrations on the
calibration system. Some of the H11 tertiary standards only have pre-deployment calibration results which
do not assess drift during use (CC71649, CA04505, CC105491) and other standards have calibration
results during their time in use but do not have post deployment calibrations that may help us evaluate the
drift rate for the last couple of weeks or months of use (ND46735, ND33801, CB11551, CB11090,
CA08107). A few standards exhibited an increased drift rate towards the end of their life that we did not
capture with the infrequent calibrations on H9. This change in drift behavior was observed as increasing
biases for measurements of target air tanks and daily test air flasks (see section 3.2). We have applied
offline mole fraction corrections to the flask air analysis $H_2$ results to correct for the end of use drift
increase for tertiary standards CC71649 (H11), CB11551 (H11) and CC305198 (H8), and the standards'
assignment uncertainty is larger for these time periods (Table 2).
Since August 2019, GML has used a newer analytical system (called MAGICC-3) in the flask air analysis
laboratory with a GC-HePDD (instrument code H8 and later H11) for $H_2$, new optical analyzers for $CO_2$,
$CH_4$ (CRDS, Picarro), CO and $N_2O$ (QC-TILDAS, Aerodyne), and a GC-ECD for $SF_6$. The responses of
the instruments on MAGICC-3 are calibrated at the same time using a single set of 11 standards spanning
a range of mole fractions for the six trace gasses. The MAGICC-3 standards were filled at the Niwot
Ridge standard air preparation facility. They are regularly measured on H9 against the GML $H_2$ primary
standards.
For the MAGICC-3 instrument response calibration, the eleven standards are analyzed sequentially
relative to an uncalibrated reference air tank (filled at Niwot Ridge). Air from each standard is injected 6
times alternating with the reference air. This entire sequence takes close to 17 hours. The first injection of
each standard is often biased low by about 2 ppb for $H_2$ due to timing issues at the start of each standard
sequence and only the remaining 5 injections are used to obtain the average normalized peak height
"signal" for each standard.
For $H_2$, a subset of 8 of the 11 working standards are used to calibrate the GC-HePDD response. The
time-dependent $H_2$ value assignment for each standard was derived from 8 or 9 calibration events on H9
between June 2018 and December 2022, listed in Table 3 (Table 3, SI Figures 1 and 2). We plan on
analyzing the MAGICC-3 standards 2 to 3 times a year going forward. The standards' $H_2$ assignments will
be revised as needed.
The time between calibration sequences was 2 weeks for the first 3 months of service of MAGICC-3 and





it has been increased to 4-5 weeks as we found the results to be quite stable. A reference air cylinder will
last 9 to 12 months on MAGICC-3. When the MAGICC-3 reference air cylinder is changed (pressure
close 250 psia), a new instrument calibration episode is done with the new reference air cylinder before
flask air samples are analyzed.
For the asynchronous calibration to stay valid up to 5 weeks requires the reference gas composition for
the six measured gasses to be stable between successive calibration dates. This has been true so far except
for one reference air cylinder CA04145 for which a small time dependent $H_2$ correction was applied
between instrument response calibration dates from 2019-11-06 to 2020-01-16 (see SI Figure 3 and more
details in SI section S4).

### 2.3 Calibration scale transfer quality assurance

GML target air tanks are dedicated air mixtures used for measurement quality control over multi-year
periods. Most are high pressure aluminum cylinders filled at the Niwot Ridge standard preparation
facility. The analysis of target air helps us evaluate the robustness of the calibration scale transfer, and the
consistency of measurements over time and also between different analytical systems. In a perfect
program, we should be able to reproduce a measurement result for a target air tank every time. As noted
earlier, however, the reality is more complicated as $H_2$ tends to grow with time in aluminum cylinders.
Tracking many aluminum cylinders provides a diverse history of behaviors (stable, or linear vs non-linear
drift), and aids in the understanding of similar cylinders used for calibration.

### 2.3.1 Calibration system (H9) Target air tanks

Some GML target air cylinders are used exclusively to evaluate the stability and performance of the H9
measurements. Other target air cylinders are analyzed on H9 and in the flask air analysis laboratory on the
H8 and H11 instruments to understand the scale transfer.
While $H_2$ has been increasing in most of our target air tanks, eleven H9 target air tanks have shown either
stable $H_2$ or a linear rate of increase less than 1 ppb/yr. Figure 4 shows the calibration histories for these
tanks as well as the residuals from the best fit for each tank. Table 4 has a list of these target tanks and
several others binned by linear drift rate. More details for target tanks and their trend best fit coefficients
are in SI Table S1. For each bin, the standard deviation of the residuals (differences of the H9 calibration
results minus the best fit values) is below 0.5 ppb. The standard deviation of all linearly drifting target
tanks residuals binned together is 0.4 ppb.
The regular analysis of target tanks on H9 (right after the instrument response has been calibrated against
the primary standards) is used to evaluate the robustness of the calibration scale transfer in GML. Results
for tanks with stable or very slowly drifting $H_2$ indicate that between 2008 and 2021, the scale transfer on
H9 has low uncertainty ( < 1 ppb).
We have eleven other target tanks for which the best fit to their calibration history is a quadratic function
(SI Figure 4 and SI Table S1). The standard deviation of these tanks' residuals binned together is 0.7 ppb.
The current set of H9 target air tank results show that residuals for higher mole fraction (>650 ppb) tanks
have a larger standard deviation (0.5-0.8 ppb, SI Figure 4d).





One tank (CC309852 A, fill date 2009-10-01) with a quadratic drift correction is on the lower end of the
GML calibration range with a $H_2$ mole fraction that grew from 204 ppb in 2009 to 232 ppb in 2021. The
standard deviation for this target tank residuals is 0.93 ppb. It appears that a quadratic fit does not capture
the observed growth very well. If we reject the first calibration result in 2009 and only fit the other
2011-2021 results showing the $H_2$ mole fraction increasing from 217 ppb to 232 ppb, the best fit is a still a
quadratic function but with smaller coefficients ($c1 = 1.66$ ppb and $c2 = -0.16$ ppb) and the standard
deviation of the residuals to this fit is reduced to 0.36 ppb. Similarly, other tanks that were analyzed soon
after fill and over several years show a similar rapid and large initial growth in $H_2$ (in the first 0.5-2 years
after fill). In this scenario, the residuals to a best linear or quadratic fit of the full calibration history will
be larger and will likely not capture the tank time-dependent $H_2$ assignment as accurately. For a few of the
GML standard and target air tanks, we dropped early calibration results that would bias the best fit
derivation and assignment during the time of use of the tank.
**2.3.2 Comparison of measurements of gas mixtures in cylinders with MPI-BGC**
Since 2016, the MPI-BGC GasLab has organized same tank air measurement ("MENI") comparisons
between WMO GAW partner laboratories as part of the European ICOS (Integrated Carbon Observation
System) Flask and Calibration Laboratory quality control work. In this program, three high pressure
cylinders are prepared and maintained by the MPI-BGC and sent to measurement laboratories in a round
robin loop. Two of the three cylinders had the same air mixture and showed small growth in their $H_2$ mole
fractions over time. The third cylinder contains an "unknown" new mixture for each round robin loop.
Between 2016 and 2021, the MENI cylinders came to GML three times and were analyzed two to four
times on the H9 instrument during each round robin stop (see SI Figure 5). Some results were rejected
due to poor instrument performance or the use of an alternate calibration strategy than the one used to
transfer the scale. For the ambient and blind $H_2$ MENI cylinders the retained NOAA $H_2$ results agree well
with the MPI_BGC measurements (< 1 ppb difference). For the low $H_2$ cylinder, the 2017/2018 NOAA
measurements are biased low by about 2 ppb while the March 10, 2021 result is about 2 ppb higher (SI
Figure 5c). The MENI program provides an important on-going check for the MPI X2009 $H_2$ calibration
scale transfer in GML.
**2.3.3 Flask analysis system (H8, H11) target air tanks**
Figure 5 shows the calibration histories for target air tanks used in the flask analysis laboratory between
2009 and 2022. $H_2$ increased in all the target tanks, sometimes rapidly, requiring time dependent value
assignments. These time-dependent $H_2$ assignments are derived from the best linear or quadratic fit to the
calibration results on H9. These assignments can be compared to the measurements on the flask analysis
systems to evaluate the quality of the scale transfer. It should be noted that the non-linear drift of some of
these tanks may not be well modeled by a simple quadratic function, leading to higher uncertainty in the
value assignments. This is especially true for tanks with limited calibration histories or gaps in their
calibration histories.



Three $H_2$ target air tanks were in service between 2009 and 2019 and have been used to evaluate the GML
calibration scale transfer to the MAGICC-1 and MAGICC-2 $H_2$ measurements (CC1824 (H), CB08834
(B) and CC303036 (A)). These tanks, however, exhibited rapid and large drifts and were not measured on
H9 on a regular basis making it more difficult to use them to evaluate potential biases on MAGICC-1 and
MAGICC-2 over this time period.
The target air tanks ALMX067998 (C) and CB11143 (C) entered service in 2016 and 2019 respectively
with more frequent measurements on the calibration system to better define the time dependent value
assignments. A new set of six target air tanks were prepared at the Niwot Ridge facility in late 2019 for
the MAGICC-3 system. They have been analyzed on MAGICC-3 multiple times a year but only one of
them has a $H_2$ mole fraction that remained below 700 ppb: CB10292 (B).
With the caveats that the non-linear drift in aluminum cylinders may not be well modeled by a simple
quadratic polynomial and that many of the early target tanks were under calibrated, the best polynomial fit
to the calibration records for all target air tanks give residuals smaller than 1.2 ppb. Details for the target
tanks, including the best fit coefficients and the standard deviation of residuals to the fits are in SI Table
S2. The uncertainty on the assignments is larger during extended time periods with no calibration results
especially for the 3 earlier target air tanks with a limited calibration history (CC1824 (H)) or with
calibration histories showing evidence of a change in the drift rate (CB08834 (B) and CC303036 (A), see
Figure 5).
In Figure 6, we show the differences between the target air analysis results on H8 and H11 and their
time-dependent $H_2$ assignments (based on the best fit to their calibration histories on H9 discussed above).
The differences are all within 4 ppb, however there are clearly times when there are persistent biases
between the flask analysis system(s) and the calibration system. Uncertainties on the value assignment of
the target air tanks, the value assignments and stability of the standards used to calibrate the flask analysis
systems as well as the noise in the H8 and H11 measurements all contribute to the observed differences.
Similar offsets on both flask analysis systems (for example CC1824 prior to 2012) may point to the main
uncertainty contribution being from the value assignment of the target air tank. Different patterns in the
offsets between the two flask analysis systems (for example offsets of different signs for CC303036 (A)
and CB08834 (B) on H8 and H11 in 2011-2013) suggest the offsets are due to value assignments of the
flask analysis system standards. Again, this is often due to limited calibration histories not being able to
fully map the non-linear drift in the standards. It also indicates there are times with systematic differences
(mostly < 2ppb) between the MAGGIC-1/H11 and MAGICC-2/H8 measurements in the flask records.
The full transition to the new MAGICC-3 system for flask analyses in August 2019 is indicated by the
vertical bar in Figure 6. As discussed earlier, one improvement in this new system is that $H_2$
measurements are now calibrated using a multi-point calibration curve from a suite of standards. This
makes the measurement results less sensitive to drift or value assignment error in any individual standard
since we are fitting multiple standards. These standards are used once a month and thus have much longer
lifetimes and longer calibration histories. We also now appreciate the complex $H_2$ growth patterns that can
occur in aluminum cylinders so have undertaken regular calibrations to ensure drift is tracked closely.
These changes seem to have reduced the bias observed between the flask analysis system and the
calibration system, which gives confidence that future measurements will be higher quality.



To help us monitor the $H_2$ calibration scale propagation going forward, a new target air tank in an Essex stainless steel cylinder, SX-1009237, was filled in late 2022 to augment the current target tanks. This target air tank should be stable for $H_2$ and will be used for periodic comparison between measurement systems. Analysis results on H9 and H11 in December 2022 are 526.75 and 527.15 ppb, respectively, consistent with the residuals for other target air tanks at that time.

## 3 NOAA flask air $H_2$ measurements

Close to 6000 flask air samples from the Global Cooperative Air Sampling Network are analyzed in GML every year. The network sites are chosen carefully to be representative of large scale air masses and to be able to rely on local support for sampling and shipping logistics. The reprocessing and release of the 2009-2021 $H_2$ global network flask air measurements on the MPI X2009 scale was made possible because of continued efforts to conduct and improve the $H_2$ measurements, to store all the necessary data, and to develop and update the tools for reliable and traceable reprocessing, comparison, and archiving.

### 3.1 Recapitulation of the GML flask air $H_2$ analysis system configurations since 2009

The MAGICC-2 H8 and MAGICC-1 H11 instruments started routine flask air $H_2$ analyses on November 5, 2009 and February 9, 2010 respectively. The flask air analysis results have been reprocessed using the tertiary standards or working standards time-dependent $H_2$ assignments on the MPI X2009 scale. As mentioned earlier, those flask air measurements on H8 and H11 until July 2019 relied on calibration with a single tertiary standard also used as reference to normalize the air sample chromatogram $H_2$ peak height.

After August 2019, the MAGICC-3 system uses a multi-standard instrument response calibration. For $H_2$, the instrument response curve is derived from eight working standards "known" assignments and their normalized $H_2$ peak heights, with a reference air that is not a standard. H8 was the $H_2$ instrument on MAGICC-3 until September 11, 2020 when it was replaced by H11 which has better precision. The linear fit coefficients for the MAGICC-3 H8 and H11 response curves are stored and used to calculate the $H_2$ dry air mole fraction in unknown air samples.

### 3.2 Data quality assurance and quality control

GML flask air $H_2$ measurements data quality is evaluated using results from the daily analysis of test air flask pairs and from the agreement between South Pole Observatory flasks pairs.

In this section, we first describe the flask sample collection protocol and introduce the data quality control tags used to document sample and measurement data quality issues. Then we assess the GML $H_2$ measurement short-term repeatability with statistics from test air flasks and South Pole Observatory flask pair differences. Finally, we present a preliminary estimation for the uncertainty of flask air $H_2$ measurements over 2009-2021, that includes empirical uncertainty estimates for the standards' assignments and the short-term noise of the instruments.

### 3.2.1 Flask air sample collection overview and data quality tagging


Partners in the NOAA Global Cooperative Air Sampling Network collect whole outside air samples in
glass flasks in pairs, upwind from any local sources of pollution, people and animals and away from
structures or terrain that would affect the wind flow. Two 2.5L glass flasks with two glass stopcocks with
Teflon o-rings are connected in series in a portable sampling unit (PSU) made of a rugged case, a battery,
a pump, an intake line, and a mechanism to control the pressure of the air samples. Most sampling units
include a dryer and are semi-automated, with the exception of those used at relatively dry high latitude
locations and a few other locations where a more rugged, manually operated sampling unit is required. At
most sites, the operator will carry the equipment outdoors to conduct the sampling. At a few sites, the
PSU is indoors and connected to a fixed inlet line drawing air from the outside.

Before shipment, the glass flasks are filled with synthetic air in the GML flask logistics laboratory.
During the sample collection on site, the flasks are first flushed for several minutes and then filled to a
pressure of 4 to 5 psi above ambient pressure in about 1 minute (See video:
https://gml.noaa.gov/education/intheair.html).

Air sample collection and/or measurement issues that are documented or detected and known to affect a
sample quality or an analyte measurement result are recorded with data quality control tags in our internal
database. For each flask air measurement, internal data quality control tags are translated into a simpler 3
column flag indicating if the measurement is retained or rejected for external data users. The GML flask
air samples and measurements can also have informational tags and comments, for example if another
measurement laboratory analyzed an air sample before it came to GML for analysis (see same air flask
comparisons in section 3.3).

The global network flasks are filled to target pressure of 17-20 psia, but the final fill pressure can vary by
a few psi, with some of the higher altitude sites having final pressures on the lower range typically. If an
air sample pressure is too low for the $H_2$ GC instrument on the MAGICC system, the $H_2$ measurement
result is tagged as "rejected" for low sample pressure. If $H_2$ measurements in paired flasks have a 5 ppb or
larger difference, the results for the pair are tagged as rejected. If only one member of the pair had an
obvious issue (leak, low flask air pressure), only the $H_2$ measurement for that member is tagged as
rejected. Some issues are detected by the MAGICC performance control system and are tagged
automatically. Other issues are tagged manually by scientists as part of regular data quality control
checks. Scientists also verify the validity of the automatic tags. Members of the team routinely evaluate if
follow-up actions are needed to fix an issue or reduce the chance of rejecting future samples for the same
issue.

Some sites can experience brief high-pollution episodes with measurements in both members of a pair
meeting the pair agreement criteria but also being outliers, i.e. outside of the expected long-term
variability at the site. For each site, we applied a smoothing curve fit calculation to determine the time
series mean behavior broken down in a long-term trend, a seasonal cycle, and shorter-term (hours to
weeks) variations [Thoning et al., 1989; Tans et al., 1989a]. The code is available and a link is provided
further down. Measurements that show large residuals from the fit are not representative of the typical
background air composition at the site and are tagged as outliers [Novelli et al., 1999]. Gross $H_2$ outliers
are typically "tagged" manually. We also apply a statistical filter before each data release, which works





iteratively to find and tag outlier $H_2$ measurements when their residuals to the curve fit is larger than 3 to
4 times the time series residuals' standard deviation.
**3.2.2 Test air flask analysis results**
Besides the regular analysis of target cylinders, the MAGICC flask analysis system is also routinely tested
using flasks filled with "test air" (flasks with site code "TST"). TST flasks are filled in one batch with air
from one of four high pressure aluminum cylinders (AL47-104, AL47-108, AL47-113, AL47-145),
themselves filled at the Niwot Ridge standard preparation site. SI Figure 6 shows the calibration histories
on H9 for different fills for these four test air cylinders. $H_2$ is not stable in the "test air" cylinders and for
some tank-fills, $H_2$ increased rapidly and grew beyond our calibration range upper limit of 700 ppb.
Every 2 to 3 weeks, the GML flask preparation and logistics laboratory manager fills an even number of
TST flasks (14-24) from the same test air cylinder. On typical analysis days, the MAGICC flask air
measurement sequence will start with the analysis of air from two TST flasks with the same fill date.
Global network flask air samples are analyzed at NOAA GML only during the daytime to ensure the
system operator is overseeing the full analysis cycle and minimizing the time a flask valve is open for the
analysis. This is meant to minimize the risk of losing or contaminating the air samples as many of them
are subsequently sent to the University of Colorado Boulder Stable Isotopes Laboratory for $CO_2$ and $CH_4$
isotope analyses.
Results from the TST flask pairs with the same fill date and analyzed on successive days give an
indication of the short-term repeatability of the measurements. Here, the deviations from the mean $H_2$ in
TST flasks with the same fill date are evaluated. For fill dates with a mean $H_2$ mole fraction less than 700
ppb, we calculate the differences between individual TST flask $H_2$ and the fill date mean. The standard
deviation of the TST flasks $H_2$ differences from their fill date mean is 1.39 ppb on MAGICC-2/H8
(N=872), 0.73 ppb on MAGICC-1/H11 (N=3583), 1.55 ppb on MAGICC-3/H8 (N=504) and 0.68 ppb on
MAGICC-3/H11 (N=1085), reflecting the higher measurement noise on H8.
Another diagnostic is the comparison of the TST flasks $H_2$ results and their test air cylinders'
time-dependent assignments for the dates the TST flasks were filled based on the best fit of the H9 test air
tank calibration results. This analysis is limited to the test air cylinders and fill code(s) with less than 700
ppb $H_2$ and with tank calibration results on H9 that reasonably capture the increase in $H_2$: AL47-108 (F),
AL47-113 (D,E,G), AL47-145 (F,G), AL47-104 (I). In SI Figure 7 (a-c), we show the $H_2$ differences
between the TST flask results and their test air cylinder assignments. The differences reflect noise in the
flask air measurements and uncertainties (and potentially small biases) in the test air tank-fill assigned $H_2$.
The three fills of test air cylinder AL47-113 are in the ambient range and have the most stable $H_2$ mole
fractions. The tank-fill assigned $H_2$ linear drift rate is 1 ppb/yr in fill D, null in fill E and 0.4 ppb/yr in fill
G. Table 5 shows the mean and standard deviation of the differences in $H_2$ between TST flasks and the
assigned $H_2$ in a stable or slowly drifting test air tank-fill. The biases for these subsets of TST air data are
less than 1 ppb and the standard deviation is equal or less than 1.5 ppb and is smaller for the most recent
MAGICC-3/H11 configuration which has a smaller number of data points.


### 3.2.3 South Pole Observatory: H$_2$ differences in flask pairs

NOAA GML operates four staffed atmospheric baseline observatories (https://gml.noaa.gov/obop/). The South Pole Observatory (SPO) in Antarctica and the Mauna Loa (MLO, Hawaii) observatories were built in connection with the 1957-1958 International Geophysical Year, a global effort bringing together 67 nations to study the Earth and in connection with the first launches of artificial satellites in Earth's orbit by the USA and the Soviet Union. The South Pole Observatory in Antarctica was established with support from the US National Science Foundation and NOAA. The other two observatories near Utqiaġvik, formerly Barrow, (BRW) and Samoa (SMO) were established in 1973 and 1974 respectively. H$_2$ time series for the observatories are shown in section 4.

The South Pole Observatory (site code SPO, sampling location: 89.98ºS, 24.80ºW,  2815 meters above sea level (masl)) gives scientists access to some of the "cleanest" air on Earth due to its remote location. On site, GML and partners operate in-situ measurements to monitor the atmosphere composition and properties, and whole air samples have been collected for trace gas analyses in the GML laboratories in Boulder since 1975.

All four NOAA atmospheric baseline observatories have an upwind clean air sector with no local sources of pollution (https://gml.noaa.gov/obop/). Every week, scientists on location collect discrete air samples when the near surface wind comes from the clean air sector. Two flask pairs are typically collected weekly and close in time at the four NOAA atmospheric baseline observatories using two collection methods. In method 'S' flasks are filled inside a building by tapping the air continuously pumped for analysis on an in-situ GHG measurement system. Method, 'P' (or 'G') involves using a portable sampling unit with an inlet mast and pump set up outside the building , similarly to other global network sites. The scientist doing the air sampling is involved with both sampling techniques. Weekly samples with both methods are typically conducted within minutes of each other. Both flask sampling methods give reliable results for H$_2$ at the South Pole Observatory.

Staff rotation and flask shipping to and from the South Pole Observatory happen during a limited time window during the Austral summer. While awaiting shipment, SPO flask air samples are stored in crates in a heated storage building. Every year, one large SPO flask shipment arrives in Boulder in December/January and another smaller shipment arrives in March. A year's worth of flasks is prepared and shipped to SPO during that same time window. Despite the longer storage for SPO flasks before analysis, we have not detected biases in H$_2$ measurements of those samples when compared with other high southern latitudes times series. SPO flask air H$_2$ measurements show close to a 20 ppb seasonal cycle and a ~15 ppb increase in the annual mean levels between 2010 and 2021 (Figure 7).

There is very little short-term variability in the surface air over Antarctica for long-lived GHGs, CO and H$_2$. The differences in the H$_2$ mole fractions in SPO paired samples therefore mostly reflect the short-term noise in the measurements. In Table 6 we report statistics for H$_2$ differences for the two flask sampling methods and the four measurement system configurations between 2009 and 2021 with H8 and H11. As observed for the TST flasks, measurements on H11 are less noisy than on H8, especially on the MAGICC-3 system.





### 3.2.4 Flask air H₂ uncertainty estimates

We have derived preliminary empirical uncertainty estimates for flask air $H_2$ measurements that fall in the 200-700 ppb range. For measurements on MAGICC-1 and MAGICC-2, the total uncertainty estimate comes from the combination of two uncertainties added in quadrature: 1) the uncertainty on the $H_2$ tertiary standard time-dependent assignment (Table 2) and 2) the instrument estimated repeatability (Table 8). If an offline assignment correction is applied to take into account changes in a standard drift rate toward the end of its use, the standard assignment uncertainty is increased. The H8 and H11 instrument repeatability estimates are listed in Table 8. For now, we assume a 0.5 ppb uncertainty on the MAGICC-3 instrument response calibrated with multiple standards. On-going work will allow us to refine this last uncertainty component estimate at a later date. Typical 1-sigma uncertainties for GML flask air $H_2$ measurements are 1.2 to 1.9 ppb on MAGICC-1, 1.4 to 2.8 ppb on MAGICC-2, 1.6 ppb on MAGICC-3/H8 and 0.8 ppb on MAGICC-3/H11.

### 3.3 Comparison with other GAW laboratories H₂ measurements

A small number of laboratories operate well-calibrated long-term measurements of important atmospheric trace gasses. The WMO Global Atmospheric Watch (GAW) coordinates regular technical and scientific discussions with experts from these laboratories. Another important outcome of the WMO/GAW collaborations consists of routine comparisons to assess the data compatibility for measurements coming from different laboratories and programs [Francey et al., 1999; Masarie et al., 2001; Jordan and Steinberg, 2011; Worthy et al., 2023]. The WMO/GAW network compatibility goals for measurements of $H_2$ in well mixed background air is 2 ppb (see Table 1 in [WMO/GAW, 2020]). This means that for $H_2$ measurement records should have persistent biases less than 2 ppb to be used in combination with other qualifying measurements in global budget, trend and large scale gradient analyses.

GML participates in several WMO GAW measurement comparison efforts. Same-flask air measurement comparisons consist of one member of a NOAA flask pair collected at a site being analyzed by a partner laboratory before being analyzed by GML. Co-located flask air measurement comparisons involve 2 or more measurement programs having samples collected at the same location and close in time. Historically, these and other "intercomparison" projects have been abbreviated ICPs, which we use in the text below. Here the GML flask air $H_2$ measurements data compatibility is assessed with results from on-going ICPs.

GML conducts same-flask air measurement comparisons at the Cape Grim Observatory (CGO, 40.68º S, 144.69º W,164 masl) with CSIRO, Australia and at the Ochsenkopf mountain top tower (OXK, 50.03º N, 11.81º E, 1085 masl) with MPI/BGC, Germany. Sampling at OXK was temporarily suspended between June 2019 and April 2021. The Alert/Dr Neil Trivett Observatory (ALT, 82.45º N  -62.51º W,  190 masl) has facilitated the largest multi-laboratory flask air comparison experiment in the WMO GAW program [Worthy et al., 2023]. NOAA has colocated flask air samples from ALT with CSIRO and the MPI/BGC. The CSIRO and MPI/BGC $H_2$ measurements are also traceable to the MPI X2009 calibration scale.

In Table 7, we summarize the annual mean of the differences for $H_2$ measurements from different laboratory and flask combinations (same flask, same flask pair or collocated flasks) for CGO, OXK and



ALT between 2010 and 2021. Columns 2 and 3 show the annual means of the NOAA $H_2$ differences between the ICP flask and its pair mate at CGO and OXK. All measurements included in the comparisons are retained, meaning they have passed quality control checks.

For CGO flask air samples collected before 2019, we find that the NOAA analysis for the NOAA ICP flask first measured at CSIRO often shows higher $H_2$ than in the non-ICP flask air sample. We suspect several of these ICP flasks had a small but detectable contamination for $H_2$. We have applied a rejection tag to NOAA analysis results for CGO ICP flasks with an $H_2$ mole fraction 2 ppb or more above $H_2$ in the non-ICP pair mate. This affected 165 ICP samples between 2009 and 2018 or 37% of all CGO ICP flasks collected between August 2009 and the end of 2021.

For OXK, the NOAA analysis result for the ICP flask first measured at MPI/BGC often shows slightly higher $H_2$ than in the non-ICP flask (Table 7, 3rd column), and the annual mean bias is less than 1 ppb for all years.

The last 4 columns in Table 7 show interlaboratory $H_2$ measurement comparisons for CGO, OXK and ALT flask air samples. The annual mean differences are consistently less than 1.6 ppb for CGO (CSIRO ICP flask and NOAA non ICP flask) and less than 2 ppb for OXK for 9 out of 11 years (MPI/BGC ICP flask and NOAA ICP flask) (Figure 8). For colocated air samples at ALT (NOAA vs CSIRO and NOAA vs MPI/BGC) we compare the mean of flask results for each laboratory and limit the comparison for samples collected within 60 minutes of each other. The ALT annual mean differences vary from year to year, are less than +/- 2 ppb for 8 years out of 12 for the NOAA vs CSIRO comparison and for 7 years out of 10 for the NOAA vs MPI/BGC comparison. These on-going ICPs are monitored regularly to continually assess the NOAA $H_2$ data compatibility with data from GAW partners.

## 4. NOAA $H_2$ atmospheric time series

Previous measurement studies have described key features of the $H_2$ global distribution for different time periods [Khalil and Rasmussen, 1990; Novelli et al., 1999; Langenfelds et al., 2002; Price et al., 2007; Yver et al., 2011]. Some of the spatiotemporal features in the more recent NOAA $H_2$ measurements at background sites are described in this section.

### 4.1 $H_2$ at NOAA Baseline Atmospheric Observatories

Figure 9 shows the reprocessed $H_2$ time series for the four NOAA Baseline Atmospheric Observatories between 2009 and 2021. Valid "S" and "P" method flask air $H_2$ measurements are retained for the South Pole Observatory only. The "S" method flasks show contaminated $H_2$ at Samoa and show some contamination or more variable $H_2$ at Utqiaġvik (Barrow). The Mauna Loa $H_2$ in "S" method flasks will be further evaluated and may be retained in future releases.

The Samoa and South Pole flask air measurements show similar maximum levels between 550 and 570 ppb and slightly higher minima at Samoa compared to the South Pole. The seasonal maxima occurs about 3 months earlier at Samoa than at the South Pole. The interannual variability is similar at both sites and is dominated by step increases on three occasions: in 2012/2013, 2016 and 2020.





The Mauna Loa $H_2$ time series shows more short-term variability reflecting the variable latitudes covered by an air mass before it is sampled at the high-altitude observatory and the strong spatial gradients for $H_2$ in the NH. The seasonal cycle amplitude at Mauna Loa is about 40 ppb with maximum levels in April-May and minimum levels in December-January. The observed seasonal maxima range from 550 to 580 ppb and the observed seasonal minima range from 505 to 520 ppb. The NOAA measurements indicate that annual mean $H_2$ levels at Mauna Loa after 2018 were higher than in previous years.

Of the four observatories, the Barrow $H_2$ time series shows the lowest levels and the strongest seasonal cycle, about 60 ppb. The observed seasonal maximum ranges from 520 to 540 ppb in April-May and the observed seasonal minimum in September-November ranges from 450 to 490 ppb.

Despite having larger emissions in the NH, the $H_2$ interhemispheric gradient shows lower levels in the extratropical NH. This is related to the larger land masses in the NH and the soil sink being the dominant removal process for $H_2$. Warwick et al. [2022] report model-based estimates for the $H_2$ lifetime of 8.3 years for the OH sink (from the authors base model configuration) and of 2.5 years for the soil uptake (average of existing literature studies). In their flux inversion, Yver et al. [2011] estimated that the NH high latitudes and the tropics represent 40% and 55% of the global soil sink respectively. The soil sink and OH sink in extratropical northern latitudes both peak in summertime [Price et al., 2007] leading to the observed stronger $H_2$ minima.

Given the larger variability and stronger seasonality of $H_2$ in the NH extra tropics, it is important to look at data from multiple sites to study and detect interannual and potentially long-term large-scale changes in atmospheric $H_2$ levels. In the next section, we highlight a few features in the global network $H_2$ records and present background air zonal mean $H_2$ time series based on samples collected at marine boundary layer sites.

## 4.2 $H_2$ at the NOAA Global Cooperative Air Sampling Network Sites

There are 51 sites considered active or recently terminated in the Global Cooperative Air Sampling Network (see map in SI Figure 8). The $H_2$ measurement times series for these sites are shown in SI Figure 9. Note that a few sites that have been discontinued are not shown in this figure. A curve fit is run for each site time series based on Thoning et al. [1989]. First the code optimizes parameters for a function made of a four-term harmonic and a cubic polynomial. The resulting residuals (measurements minus function) are then smoothed with a low-pass filter with a 667-day cutoff and are added to the polynomial part of the function to produce the "trend curve" (shown as the dark blue line in site time series plots in SI Figure 9). The residuals are also smoothed with a low-pass filter with a 80 day cutoff and are added to the function to produce a "smooth curve" at each site.

The data quality control work on our long-term measurement time series includes a data selection step [Dlugokencky et al., 1994]. Samples with $H_2$ beyond 3.5 (4 for ASC) standard deviations of the time series smoothed curve at each site are flagged as not representative of background air conditions and are shown as crosses in SI Figure 9.

The annual mean, maximum and minimum $H_2$ values of the smooth curve for the 51 sites are plotted in Figure 10 (in order of decreasing latitude along the x-axis) for years with retained measurements up to





2021. Sampling at the TPI site, on Taiping Island, Taiwan, started in May 2019, which explains the 2 (full
sampling year) data points for the site. Sampling at a few network sites was impacted by the covid-19
pandemic resulting in data gaps or delayed return shipping of samples. We recommend data users become
familiar with individual sampling site measurement records to best aggregate and interpret signals.
The interhemispheric gradient of $H_2$, with higher levels in the SH, is apparent in the annual means
distribution across sites in Figure 10 (green circles). The majority of sites in the SH (BKT to SPO on the
right side of Figure 10) show smaller seasonal cycle amplitudes (<23 ppb) than NH sites; however,
several sites have interannual variations in their $H_2$ seasonal cycle amplitudes (SI Figure 9). Sites with the
lowest $H_2$ seasonal minima (Figure 10, blue x symbols) likely are the most influenced by soil uptake. A
few sites (for ex. TAP, AMY, LLN, CPT) show higher smooth curve annual maxima (Figure 10, red
crosses), likely reflecting upwind local or regional emissions.
Two sites, KUM and WIS, had a change of sampling location that resulted in visibly different $H_2$ mean
levels and seasonal cycle amplitudes. In mid 2018, the KUM site was moved 30 km NNW along the
Hawaii island SE coastline when access to a lava field bordering the ocean was lost in the eruption of the
Kīlauea volcano. The KUM sampling location change resulted in higher mean $H_2$ levels and a smaller
seasonal cycle. The WIS site moved 100 km SSE in Israel in early 2015. There are more instances of
depleted $H_2$ (in December-March) since the move, potentially reflecting a stronger influence of soil
uptake in air masses sampled at the newer location.

**835 4.3 $H_2$ marine boundary layer global and zonal means**

To extract large scale signals from the global air sampling network, we use the NOAA GML marine
boundary layer (MBL) zonal data product [Tans et al., 1989b; Dlugokencky et al., 1994]. Time series
from remote MBL sites are smoothed and interpolated to produce a latitude versus time surface of the $H_2$
mean MBL mole fraction (Figure 11). For $H_2$, the number of sites included in the zonal mean calculations
ranges from 29-42 sites until July 2017 when sampling from the Pacific Ocean shipboard (POC) was
stopped, after which 24-27 sites were included in the calculation. Because the Global Cooperative Air
Sampling Network is sparse in the tropics and in the SH mid latitudes, the MBL product likely does not
equally detect and reflect interannual variability in fluxes in these under-sampled regions, for example
biomass burning emissions in Africa and South America.
To further isolate changes in background $H_2$ at different latitudes, we first calculate MBL global (and
zonal band) means (shown in SI Figure 10) and then derive anomalies by removing the 2010-2021
average year from the global (or zonal band mean) time series. Figure 12 shows the MBL anomaly for $H_2$
(black lines) and CO (dashed blue lines) for the global mean and 5 zonal band means (NH and SH Polar
(53-90º), NH and SH Temperate (17.5-53º) and Tropics (17.5ºS to 17.5ºN). The NOAA GML CO
measurements are for the same air samples as the $H_2$ measurements [Pétron et al., 2023b]. Here, we derive
the global and zonal means for CO using the 2009-2022 MBL CO measurements and the anomalies are
based on the 2010-2021 smooth curve zonal mean results to be consistent with the $H_2$ data analysis.
CO is emitted during incomplete combustion and is a useful marker of biomass burning emissions. CO
has a shorter atmospheric lifetime than $H_2$ which results in shorter-lived CO anomalies from pulse
emissions. The data reduction for the anomaly analysis is slightly different from Langenfelds et al. [2002]





investigation of $CO_2$, $CH_4$, $H_2$, and CO interannual variability in the CSIRO network 1992-1999 time
records. The CSIRO authors employed the same [Thoning et al., 1989] data smoothing technique as we
do but used the derivative of the trend curve to analyze correlations in interannual growth rate variations
between species. The anomaly approach chosen here allows to retain the timing of abrupt changes in the
measurement records.
Over 2010-2021, background air $H_2$ has increased at all latitudes (Figure 12). The global mean MBL $H_2$
shows a non-uniform increase over this time with a noticeable 10 ppb step increase in 2016. The global
mean MBL $H_2$ was 20.2 ±0.2 ppb higher in 2021 compared to 2010 (Figure 12a).
The meridional gradient and zonal band mean plots (Figure 11 and Figure 12b-f) highlight the evolution
of background air $H_2$ at different latitudes. By construction, the smooth curve anomalies are not directly
proportional to the biomass burning emissions that likely caused them. Rather the anomalies in the
smooth curves are useful to point to time periods when several successive air samples at a site show
similar deviations from the average seasonal cycle and multi-year trend.
The 2016 $H_2$ step increase is detected in the Tropics and SH. In the Tropics it coincides with a strong
positive CO anomaly that started in November 2015, reached a peak amplitude of 15 ppb mid-January
2016 and ended in May 2016. The 2015/2016 $H_2$ anomaly is first detected at Bukit Kototabang, Indonesia
(BKT) and later at Ascension Island (ASC), Cape Grim (CGO) and Crozet Island (CRZ) (SI Figure 11).
Some BKT air samples impacted by biomass burning emissions show enhancements of 100s ppb in CO
and $H_2$. The BKT CO and $H_2$ data also show enhancements likely related to biomass burning in 2015. The
2015 fire season in Indonesia was among the most intense on record as shown by remote sensing products
of fire counts, CO and aerosols. Field et al. [2016] found that burning activities to clear peatland for
farming likely contributed to larger emissions than expected from dry conditions alone in 2015.
There is another step increase in the Polar SH zonal band in early 2020, also coinciding with a pulse
anomaly in CO (Figure 12f) likely related to large wildfires in Australia in late 2019-early 2020. The
Cape Grim (CGO) and Crozet Island (CRZ) smoothed curves show a large jump between the late 2019
minimum and early 2020 maximum when the CGO CO measurement seasonal minimum is also 10-12
ppb higher than in other years (SI Figure 11). van der Welde et al. [2021] estimate that the 2019-2020
fires in Australia emitted 80% more $CO_2$ than "normal" Australian annual fire and fossil fuel emissions
combined.
In the NH extratropics bands, positive anomalies in $H_2$ in 2021 coincide with CO pulse anomalies. For the
Polar (Temperate) NH zonal band, the CO anomaly lasts from mid-July (June) to December 2021 with a
peak in September and an anomaly maximum amplitude of 37 ppb (19 ppb). Record high emissions of
$CO_2$ and CO from boreal forest fires in Eurasia and North America in 2021 have been reported by Zheng
et al. [2023].
Previously, Simmonds et al. [2005] and Grant et al. [2010] have reported on the observed variability in
the Mace Head continuous $H_2$ measurement record and linked interannual variability in the baseline
annual mean $H_2$ to larger fire emission events. More recently, Derwent et al. [2023] shared an updated
analysis of the February 1994-September 2022 Mace Head in-situ $H_2$ measurements. The in situ record
shows higher monthly mean baseline $H_2$ levels in recent years and the authors report an increase in





monthly mean anomalies after December 2015 (slope of 2.4 +/- 0.5 ppb/yr). They postulate that a
"missing" source of increasing intensity after 2010 may be behind the observed sustained increased $H_2$,
which is markedly different from the 1998-1999 anomalies attributed to biomass burning. Derwent et al.
[2023] explore potential candidates for the missing sources. However, in the absence of strong and
quantitative direct evidence at this time, additional studies are needed to interpret the observed $H_2$
variability.
**5. Conclusions**
In this paper, we have described how NOAA GML has adopted the MPI X2009 $H_2$ calibration scale. The
work was confined to measurements on GC-HePDD instruments. The GML $H_2$ primary standards in
electropolished stainless steel cylinders have been calibrated once by the MPI CCL in Fall 2020. We have
used the CCL assignments to propagate the scale to secondary and tertiary standards. $H_2$ increases in
most air standards stored in aluminum cylinders. A curve fit is applied to each standard calibration history
to determine a time-dependent $H_2$ assignment on MPI X2009. The tertiary and working standards $H_2$
assignments were then used to reprocess results for NOAA flask air $H_2$ measurements on MPI X2009.
These NOAA Global Cooperative Air Sampling Network flask reprocessed $H_2$ measurements for
2009-2021 are now publicly available [Pétron et al., 2023a]. For the period 2010-2021, same air
measurements with GAW partner laboratories have annual mean differences less than 2 ppb for the Cape
Grim comparison with CSIRO and less than 3 ppb for the Ochsenkopf comparison with MPI BGC. Over
2010-2021, background air $H_2$ has increased at all latitudes. However, site time series and marine
boundary layer $H_2$ zonal means show significant interannual variability. We find that some of strongest $H_2$
zonal mean anomalies coincide with CO anomalies and therefore were likely driven by large biomass
burning events in Indonesia (2015), Australia (2019/2020), and boreal latitudes (2012 and 2021) [Field et
al., 2016; Petetin et al., 2018; Zheng et al., 2023]. A full analysis of the NOAA Global Cooperative Air
Sampling Network $H_2$ measurement records is beyond the scope of this paper. This dataset complements
WMO/GAW partner laboratories $H_2$ measurements and it will be updated and extended routinely
moving forward.

**Data and Code Availability**
The NOAA global network flask air $H_2$ and CO time series are available at
https://doi.org/10.15138/WP0W-EZ08.
We kindly request that users of the NOAA $H_2$ dataset cite:
Pétron, G., Crotwell, A., Crotwell, M., Kitzis, D., Madronich, M.,
Mefford, T., Moglia, E., Mund, J., Neff, D., Thoning, K., & Wolter, S.
(2023). Atmospheric Hydrogen Dry Air Mole Fractions from the NOAA GML Carbon
Cycle Cooperative Global Air Sampling Network, 2009-2021 [Data set].
NOAA GML CCGG Division. Version: 2023-05-25, https://doi.org/10.15138/WP0W-EZ08
The python class used to filter and smooth time series data is available and explained at:
https://gml.noaa.gov/aftp/user/thoning/ccgcrv/ccgfilt.pdf and the method can be referenced as
[Thoning et al., 1989].



**Supplement**

The supplement for this article is available in a separate file.

**Author Contributions**

GP and AC designed the scale revision work. GP, AC and JM implemented the scale revision. GP, AC, MC, MM, DN and JM contributed to the data quality control. GP and JP analyzed network site time series. AC designed, built and oversaw the $H_2$ calibration scale transfer and the flask air analysis system operations, working with Paul Novelli until he retired in 2017. TM and AC carried out tank calibrations. BH prepared the primary standards. DK was in charge of the whole air secondary and tertiary standards preparation. MM and EM were responsible for the flask air analysis lab operations, working with Patricia Lang until her retirement in 2019. EM managed the flask logistics laboratory and flask metadata entries. DN with support from SW managed the NOAA Global Air Sampling Network. DN managed sampling equipment for sites. JM manages the database and date releases, JM, KT and AC developed code and user interfaces for data processing, quality control and exploration. AJ calibrated the NOAA primary standards. AJ, PK and RL contributed data from their measurement programs. GP prepared the manuscript with contributions from AC and AJ and edits from BH, MC, RL, and JP.

**Competing Interests**

The authors declare that they have no conflict of interest.

**Acknowledgements**

We are grateful for our partners worldwide who collect and ship flask air samples to NOAA GML, Boulder, CO for analysis. We thank past and current NOAA GML and CU CIRES colleagues for their contributions to the network operations, measurements, data management and data quality control. Gary Morris and Kathryn McKain provided valuable comments on the manuscript.

**Financial support**

This research was supported in part by NOAA cooperative agreements NA17OAR4320101 and NA22OAR4320151.



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



## Tables

Table 1. NOAA GML $H_2$ primary standards (prepared gravimetrically) and their WMO/MPI X2009
assignments. All $H_2$ dry air mole fractions and their uncertainties are in ppb.

| Serial Number | Fill code | Fill Date | CCL value | CCL uncertainty |
|---|---|---|---|---|
| SX-3558 | A | 2008-10-17 | 248.4 | 0.1 |
| SX-0614470 | A | 2019-04-15 | 352.8 | 0.1 |
| SX-3543 | B | 2008-11-03 | 425.4 | 0.2 |
| SX-3540 | B | 2007-08-07 | 488 | 0.2 |
| SX-0614471 | A | 2019-04-19 | 496.5 | 0.3 |
| SX-3523 | C | 2007-07-24 | 527 | 0.2 |
| SX-3554 | A | 2007-08-02 | 601.2 | 0.2 |
| SX-0614472 | A | 2019-04-19 | 701.9 | 0.2 |

Table 2: $H_2$ secondary standards used in the tank calibration laboratory and $H_2$ tertiary standards used on
the MAGICC-1 and MAGICC-2 systems (2009 to 2019).

| Tank Calibration / H9 | | | | | | | | |
|---|---|---|---|---|---|---|---|---|
| Tank ID (fill) | Time of use | t0 | Assignment at t0 (ppb) | C1 (ppb/yr) | C2 (ppb/yr$^2$) | N | Residuals standard deviation (ppb) | Fill date |
| CC119811 (A) | 2/05/2008 to 6/02/2013 | 2010.0689 | 549.4 | 2.0 | 0 | 47 | 0.50 | 2006-01-01 SM* |
| CA03233 (B) | 6/02/2013 to 11/01/2018 | 2016.7106 | 502.8 | 0 | 0 | 19 | 0.23 | 2010-08-12 NWR |
| MAGICC-1 / H11 | | | | | | | | |
| Tank ID (fill) | Time of use | t0 | Assignment at t0 (ppb) | C1 (ppb/yr) | C2 (ppb/yr$^2$) | N | Assignment uncertainty (ppb)** | Fill date |
| CA08107 (D) | 7/22 to 8/7/2019 | 2019.2959 | 562.9 | 15.4 | 0 | 6 | 0.6 | 2018-11-09 NWR |
| CB11090 (B) | 10/18/2018 to 7/19/2019 | 2019.1482 | 576.3 | 6.9 | 0 | 4a | 0.6 After 2019-06-21: 1.5 | 2016-9-30 NWR |
| CB11551 (A) | 2/13 to 10/17/2018 | 2018.1878 | 548.8 | 6.7 | 0 | 3a,b, c | 0.5 After 2018-08-27: 1.5 | 2015-1-01 SM* |
| CC91285 (C) | 6/19/2017 to 2/13/2018 | 2017.1711 | 538.4 | 0 | 0 | 8 | 0.5 | 2015-8-14 NWR |
| CA08165 | 10/13/2016 to | 2016.9137 | 535.7 | 4.5 | 0 | 3c | 0.5 | 2011-12-16 |





| | | | | | | | | |
|---|---|---|---|---|---|---|---|---|
| (B) | 06/16/2017 | | | | | | | NWR |
| CC302566 (B) | 3/21/2016 to 10/12/2016 | 2016.3645 | 540.2 | 4.4 | 0 | 5 | 0.5 | 2015-8-14 NWR |
| CC105491 (B) | 8/10/2015 to 3/18/2016 | 2015.1506 | 522.3 | 0 | 0 | 5d | 1.0 | 2014-1-16 NWR |
| ND33801 (B) | 8/4/2014 to 8/7/2015 | 2013.8771 | 509.3 | 0.9 | 0 | 6e | 0.5 After 2015-05-14: 1.0 | 2012-12-27 NWR |
| CB09117 (A) | 2/18 to 8/1/ 2014 | 2013.8912 | 635.3 | 28.7 | 0 | 5 | 2 | 2012-12-17 SM |
| ND46735 (A) | 9/10/2012 to 2/13/2014 | 2012.9158 | 527.4 | 2.5 | -1.0 | 7e,f | 0.5 After 2013-12-11: 1.0 | 2011-1-1 estimated |
| CA04505 (B) | 12/9/2011 to 9/7/2012 | 2011.4593 | 540.6 | 1.7 | 0 | 3c,d | 1.0 | 2010-8-12 NWR |
| ND38963 (A) | 8/12/2010 to 12/7/2011 | 2011.704 | 586.0 | 6.2 | 0 | 4 | 0.5 | 2009-1-1 estimated |
| CC71649 (E) | 1/22 to 8/6 2010 | 2009.1184 | 507.1 | 8.4 | 0 | 7b,e | 1.5 | 2008-9-19 |
| MAGICC-2 / H8 | | | | | | | | |
| Tank ID (fill) | Time of use | | Assignment at t0 (ppb) | C1 | C2 | N | Assignment uncertainty (ppb)** | Fill date |
| ND38954 (B) | 3/26/2013 to 3/21/2014 | 2014.2094 | 516.6 | 2.0 | 0 | 5 | 0.5 | 2012-12-9 NWR |
| CA03409 (B) | 5/23/2011 to 3/25/2013 | 2011.6278 | 526.6 | 0 | 0 | 5e | 0.5 After 2013-01-21: 1.0 | 2010-1-1 estimated |
| ND38415 (A) | 4/5/2010 to 5/20/2011 | 2010.2502 | 566.1 | 20.9 | -8.7 | 6 | 0.5 | 2009-1-1 estimated |
| CC305198 (A) | 11/2//2009 to 4/3/2010 | 2009.7211 | 557.9 | 65.8 | 0 | 3a,b | 1.5 After 2010-01-31: 2.5 | 2009-1-1 SM* |

* Gravimetric blends with CO, $H_2$, $CO_2$, $CH_4$ and $N_2O$ in zero air purchases from Scott Marrin.
** Uncertainty estimates listed for the tertiary standard assignments assume a 0.5 ppb uncertainty for each
calibration result on H9 and do not formally include the uncertainty on the secondary standard
assignments.
a. Assignment does not use existing post-use calibration results that show larger drift
b. Drift change towards end of use, additional drift correction applied.
c. Force linear fit in drift calculation code
d. Only predeployment calibrations
e. No end-of-use or post-use calibration
f. Force quadratic fit in drift calculation code





Table 3: H$_2$ working standards used on the MAGICC-3 system. Best polynomial curve fit coefficients to
the August 2019-December 2022 calibration histories.

| Tank ID (fill) | t0 | Assignment at t0 (ppb) | C1 (ppb/yr) | C2 (ppb/yr$^2$) | N | Assignment uncertainty (ppb) | Fill date |
|---|---|---|---|---|---|---|---|
| CA01414 (I) | 2020.0964 | 238.4 | 10.0 | -1.9 | 9 | 0.5 ppb | 2017-12-29 NWR |
| CA04403 (F) | 2020.1052 | 474.6 | 10.2 | -1.7 | 9 | 0.5 ppb | 2017–12-1 NWR |
| CB11270 (A) | 2020.0012 | 515.0 | 2.9 | -0.5 | 9 | 0.5 ppb | 2017-12-1 NWR |
| CA06388 (H) | 2019.9423 | 551.2 | 1.1 | 0 | 9 | 0.5 ppb | 2018-2-23 NWR |
| CA05773 (F) | 2020.2585 | 565.6 | 1.4 | 0 | 8 | 0.5 ppb | 2018-5-17 NWR |
| CB11034 (B) | 2020.0783 | 580.1 | 8.3 | -1.2 | 9 | 0.5 ppb | 2018-5-17 NWR |
| CA05680 (H) | 2020.0904 | 588.1 | 1.9 | 0 | 9 | 0.5 ppb | 2017-12-1 NWR |
| CB11405 (C) | 2020.1474 | 605.6 | 23.3 | -1.6 | 9 | 0.5 ppb | 2018-5-17 NWR |







Table 4: H9 Target air tanks with zero or linear growth in $H_2$

| Linear Drift Rate (ppb/yr) | Target Tank IDs | Standard deviation of residuals to best fits (ppb) |
|---|---|---|
| 0 | CA05278, CA06194, CA08247, CC121971, CC311842 ND16439, ND33960 | 0.46 |
| 0-1 | ALM-065166, CA05300, CC71607, CC73110 | 0.42 |
| 2-5 | CA04551, CA07328, CB10910 | 0.32 |
| 5-10 | CC71579 | 0.36 |
| > 20 | CA08145 | 0.48 |






Table 5. Summary statistics for $H_2$ differences between test air tank-fill assignment (based on H9
calibration history) and associated TST flask measurements on MAGICC systems

| System / Instrument | Test air tank id and fill | Differences mean (ppb) | Differences standard deviation (ppb) | Number of samples |
|---|---|---|---|---|
| MAGICC-2 / H8 | AL43-113 D, E | -0.3 | 1.3 | 528 |
| MAGICC-1 / H11 | AL43-113 D, E, G | +0.3 | 1.1 | 1231 |
| MAGICC-3 / H8 | AL47-145 G | -0.9 | 1.5 | 388 |
| MAGICC-3 / H11 | AL43-113 G | +0.4 | 0.6 | 144 |




Table 6. Summary statistics for SPO flask pair $H_2$ differences. Npairs= Number of flask pairs.

| System/ Instrument | SPO "P" flasks Absolute differences | | | SPO "S" flasks Absolute differences | | | SPO "S"-"P" Pair mean differences | | |
|---|---|---|---|---|---|---|---|---|---|
| | Mean (ppb) | Std dev (ppb) | Npairs | Mean (ppb) | Std dev (ppb) | Npairs | Mean (ppb) | Std dev (ppb) | Npairs |
| MAGICC-2 / H8 | 1.3 | 1.0 | 165 | 1.1 | 0.9 | 87 | -0.4 | 1.5 | 81 |
| MAGICC-1 / H11 | 0.9 | 0.8 | 292 | 0.9 | 0.8 | 143 | -0.2 | 1.3 | 144 |
| MAGICC-3 / H8 | 1.6 | 1.3 | 45 | 1.2 | 1.2 | 25 | -0.1 | 1.7 | 25 |
| MAGICC-3 / H11 | 0.7 | 0.6 | 76 | 0.8 | 0.6 | 35 | -0.5 | 0.8 | 43 |








Table 7: Annual mean of $H_2$ measurement differences for air samples from the Cape Grim Observatory
(CGO), Ochsenkopf (OXK) and Alert (ALT). Non background air sample measurement results are
included. Collocated (not same air) samples at ALT are matched within a +/- 60 minutes window.
[updated 9-25-23]

| Year | NOAA ICP-NOAA nonICP | | CGO NOAA non ICP minus CSIRO ICP | OXK NOAA ICP minus MPI ICP | ALT NOAA minus CSIRO (not same air) | ALT NOAA minus MPI (not same air) |
|---|---|---|---|---|---|---|
| | CGO* | OXK | | | | |
| 2010 | - | -0.05 | 0.72 | -0.17 | -3.4 | -3.5 |
| 2011 | - | 0.15 | 0.50 | -0.02 | 2.2 | -3.9 |
| 2012 | 0.58 | 0.13 | 0.40 | -0.29 | 0.66 | -2.3 |
| 2013 | - | 0.01 | 0.23 | 0.80 | 1.30 | -1.4 |
| 2014 | - | 0.19 | 1.37 | 1.61 | 0.63 | -1.1 |
| 2015 | - | 0.85 | 0.02 | 0.53 | 0.52 | -1.4 |
| 2016 | 1.32 | 0.20 | 1.54 | 2.91 | -0.32 | -1.4 |
| 2017 | 1.19 | 0.56 | 1.38 | 2.49 | 3.2 | - |
| 2018 | 0.91 | 0.53 | 1.31 | 1.69 | 1.2 | -1.3 |
| 2019 | 0.73 | -0.07 | 0.30 | 1.25 | 1.0 | -0.81 |
| 2020 | 0.18 | na | 0.19 | - | 0.01 | -0.22 |
| 2021 | 0.33 | 0.33 | 0.86 | 1.71 | 3.4 | - |

*Most NOAA ICP flasks from CGO had a small contamination for CO and $H_2$ prior to 2019. If the
NOAA ICP flask $H_2$ results are > 2ppb larger than the NOAA non-ICP flask $H_2$ in the pair, the ICP flask
$H_2$ has been rejected. Only years with at least 10 valid $H_2$ pairs are included.





Table 8: Flask air $H_2$ measurement uncertainty components

| Uncertainty components | 1 sigma uncertainty estimate (ppb) | Source |
|---|---|---|
| Tertiary standard time-dependent assignment uncertainty (1 point calibration) | 0.5-2.5 Tank specific (see Table 2) | Calibration histories, residuals to best fit, TST flasks |
| MAGICC-3 response curve uncertainty | 0.5 | Preliminary estimate, will be reassessed. |
| Measurement repeatability on H8 | 1.3  (MAGICC-2) 1.5 (MAGICC-3) | TST and SPO flask pair differences (Tables 5 and 6) |
| Measurement repeatability on H11 | 1.1 (MAGICC-1) 0.6 (MAGICC-3) | |






**Figures**

Figure 1. Calibration results for GML two H$_2$ secondary standards (a) CC119811 and b) CA03233) on H9 against one of the primary standards. 2019-2020 multipoint calibration results on H9 are also shown for CA03233 (pink circles). Only results shown with open circles are used for the assignments.

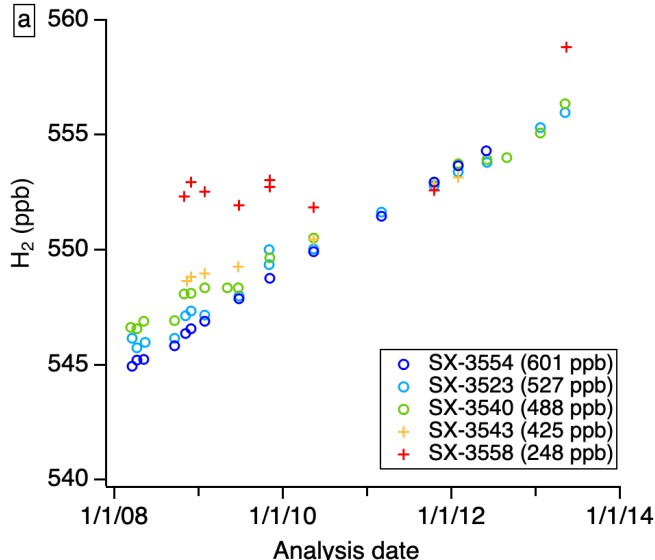

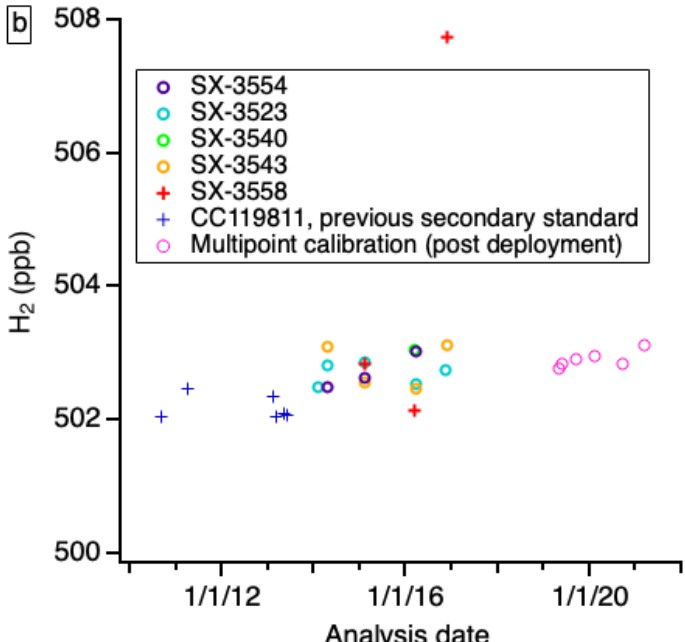





Figure 2: 2019-2022 H9 standard calibration response curve (RC) results: a) differences from the mean RC linear fit mean and b) residuals of the response curve fits. Different colors are for different calibration episodes.

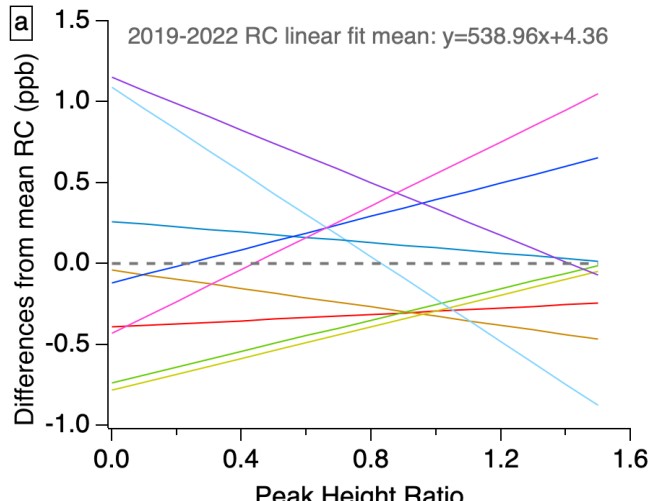

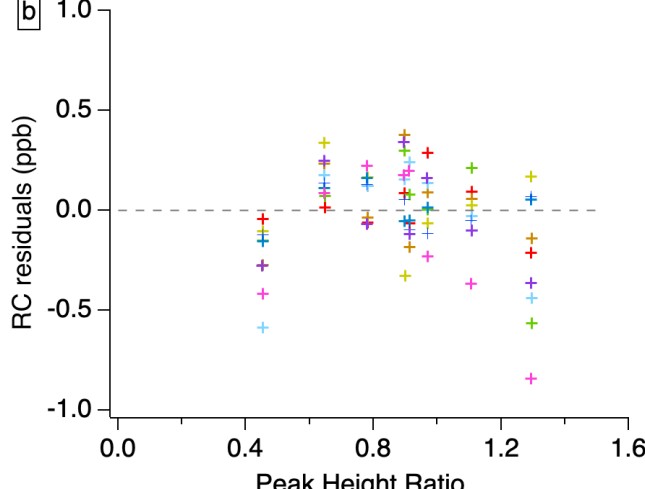



Figure 3. Calibration histories of a) MAGICC-1 / H11 and b) MAGICC-2 / H8 tertiary standards. The colored vertical line indicates when a standard started to be used.

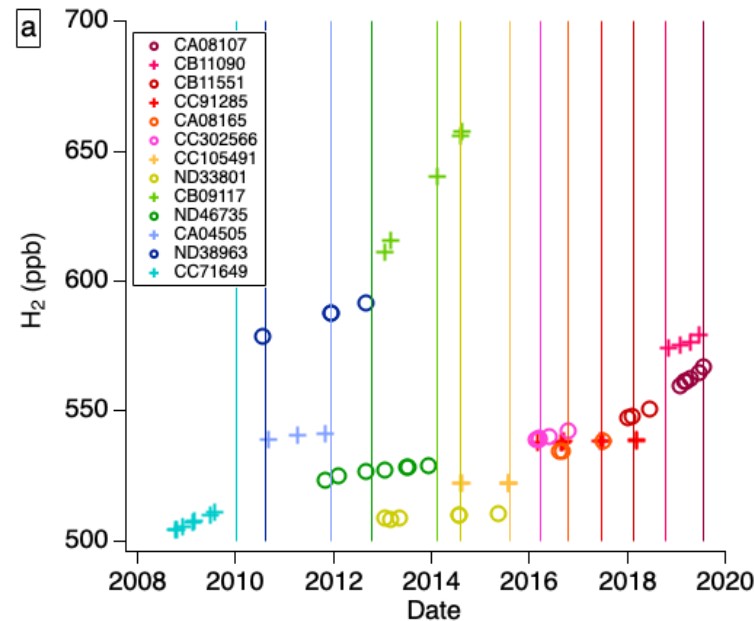

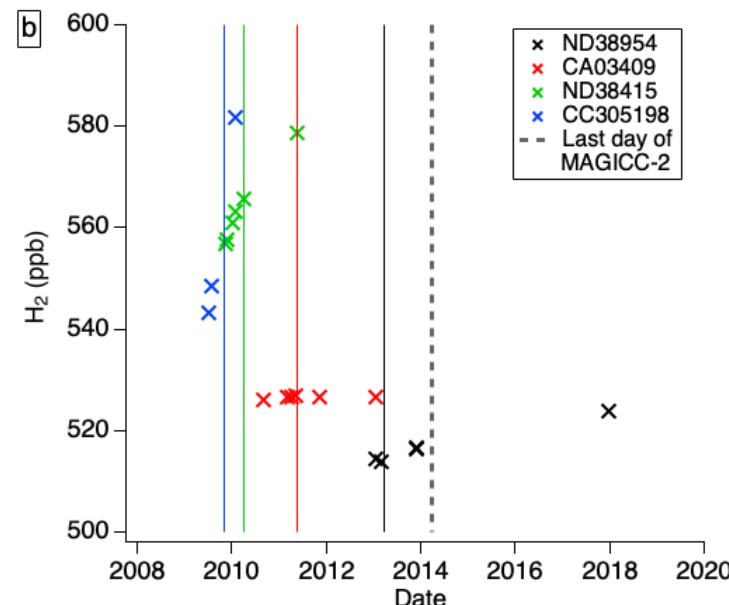





Figure 4: Calibration histories and residuals to best fit for H9 target tanks with a stable $H_2$ mole fraction
or a linear drift less than 1 ppb/yr. Residuals are in ppb.





Figure 5. Flask air analysis system (H8 and H11) target air tanks H9 calibration histories and residuals to
best linear or quadratic fit.

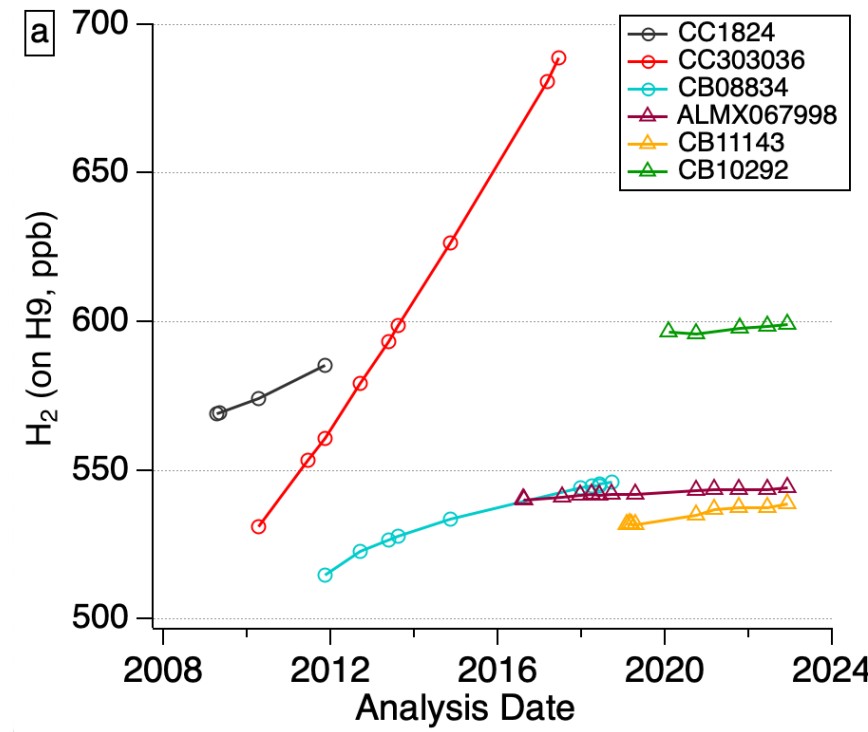







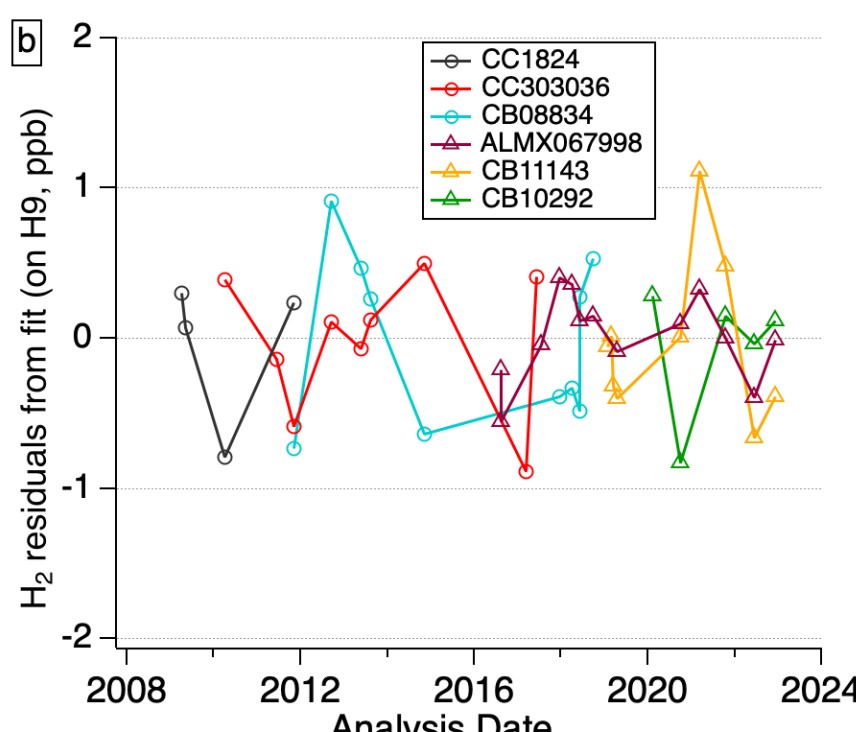




Figure 6. Differences of target air tank $H_2$ analysis results on MAGICC and time-dependent assignment
based on calibration history on H9. The vertical line indicates the transition to the MAGICC-3 flask
analysis system.

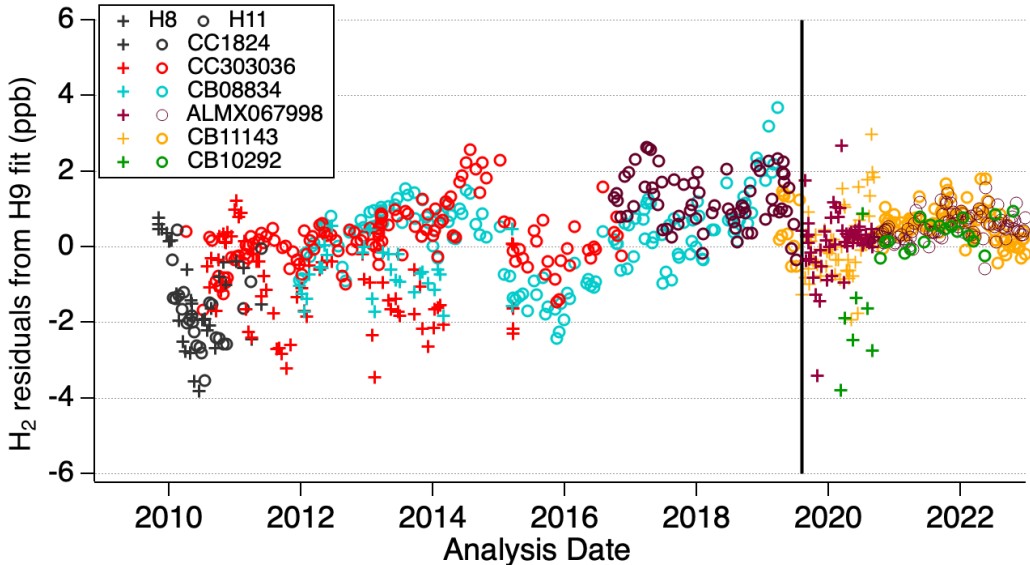



Figure 7. South Pole Observatory flask air $H_2$ measurements. Circle and "+" symbols refer to instruments:
H11 or H8. Black is used for measurements of P flasks on the MAGICC-1 or MAGICC-2 system and red
for the MAGICC-3 system.  Light green is used for measurements of S flasks on the MAGICC-1 or
MAGICC-2 system and light blue for the MAGICC-3 system.


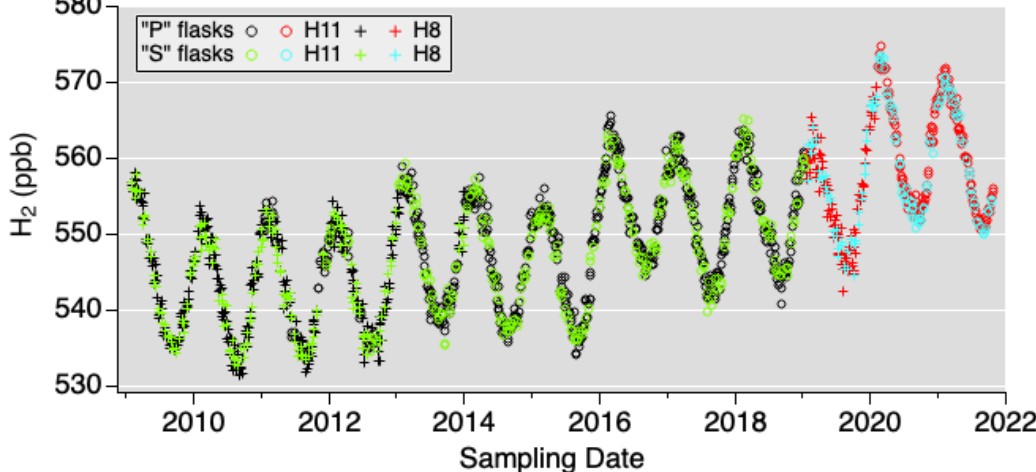






Figure 8. Interlaboratory same air H$_2$ measurement difference for OXK ICP (NOAA - MPI BGC) and CGO (NOAA non ICP - CSIRO ICP).

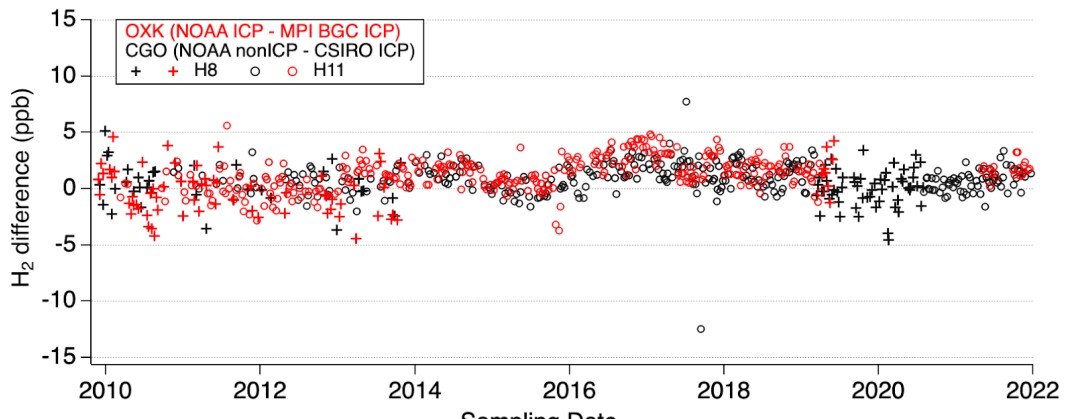

Figure 9. H$_2$ time series at the NOAA Baseline Atmospheric Observatories

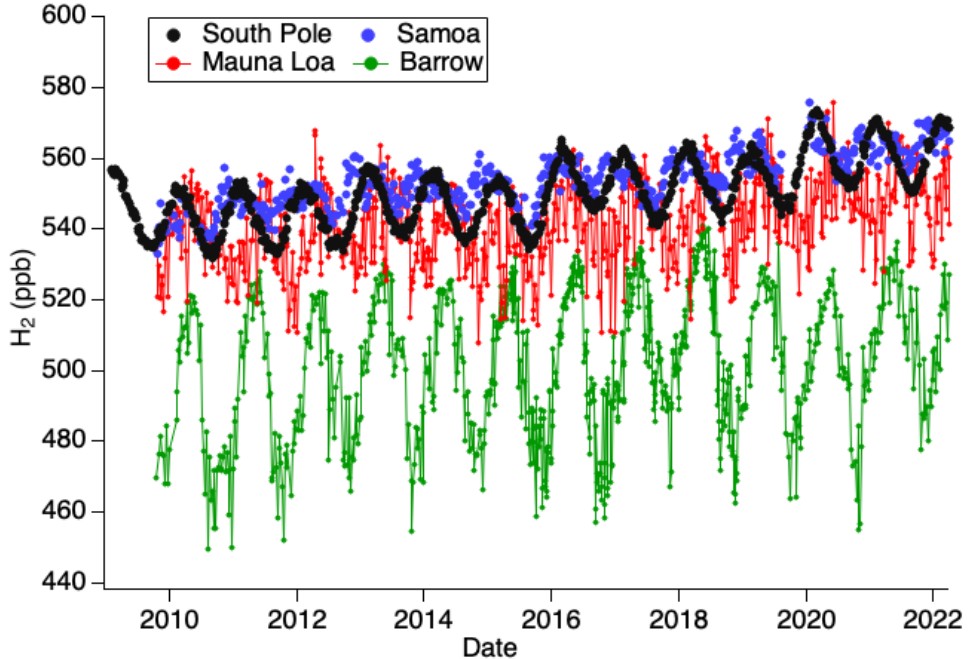



Figure 10: Annual maximum (red), mean (green) and minimum (blue) H$_2$ from the smooth curve fit of the 2010-2021 measurement time series for each surface site in the global sampling network. Each site is referred to with a three letter code (see details at https://gml.noaa.gov/dv/site/). The sampling sites are shown along the x-axis with decreasing latitudes. An asterisk near the site code indicates if the site data is used for the marine boundary layer air zonal and global means data reduction.

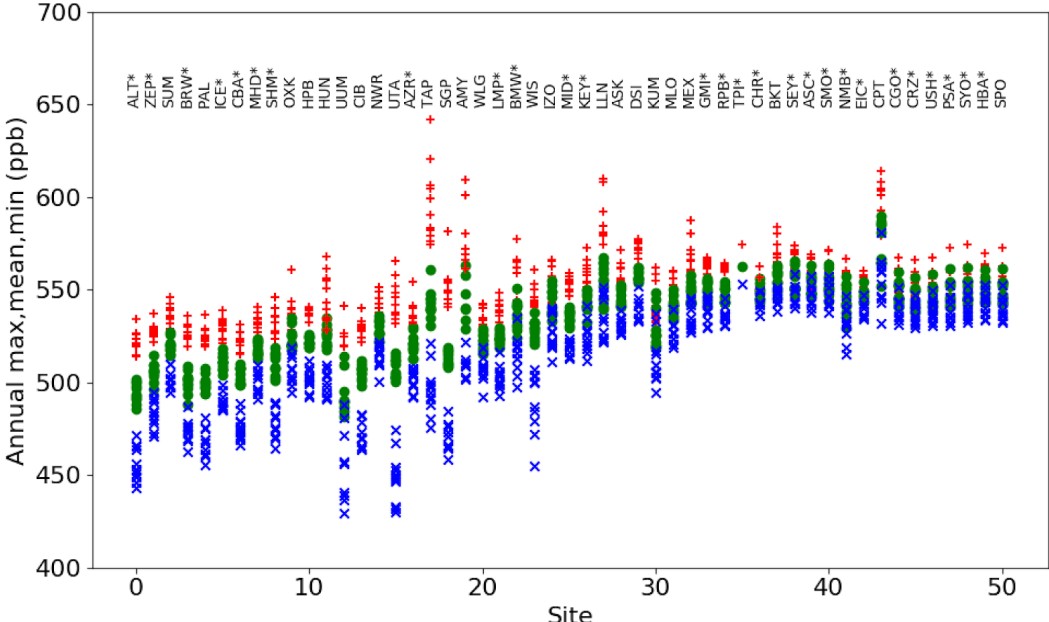



Figure 11: 2010-2021 marine boundary layer $H_2$ meridional gradient. Y-axis is the sine of latitude.

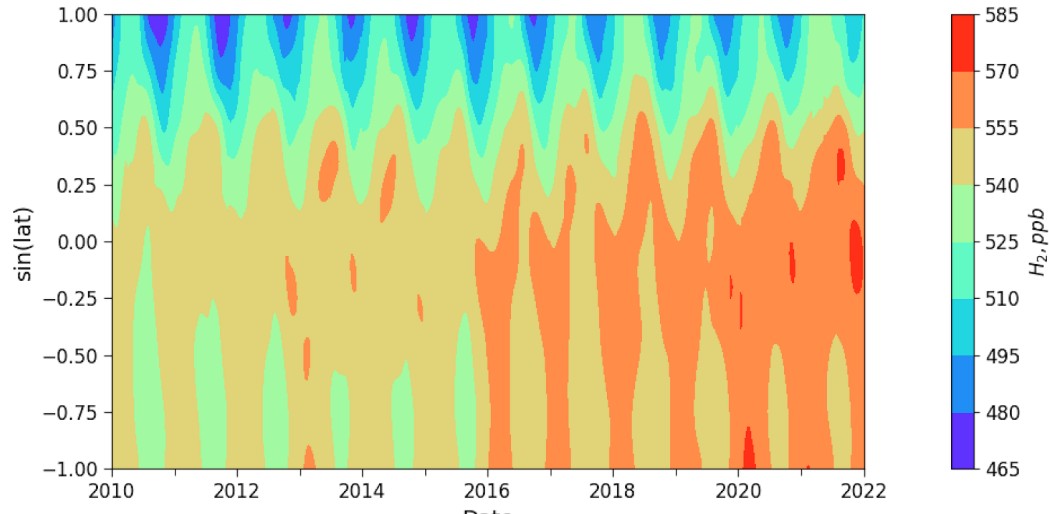








Figure 12: 2010-2021 marine boundary layer global and zonal mean $H_2$ anomaly (black line) and CO
anomaly (dashed blue line) time series.

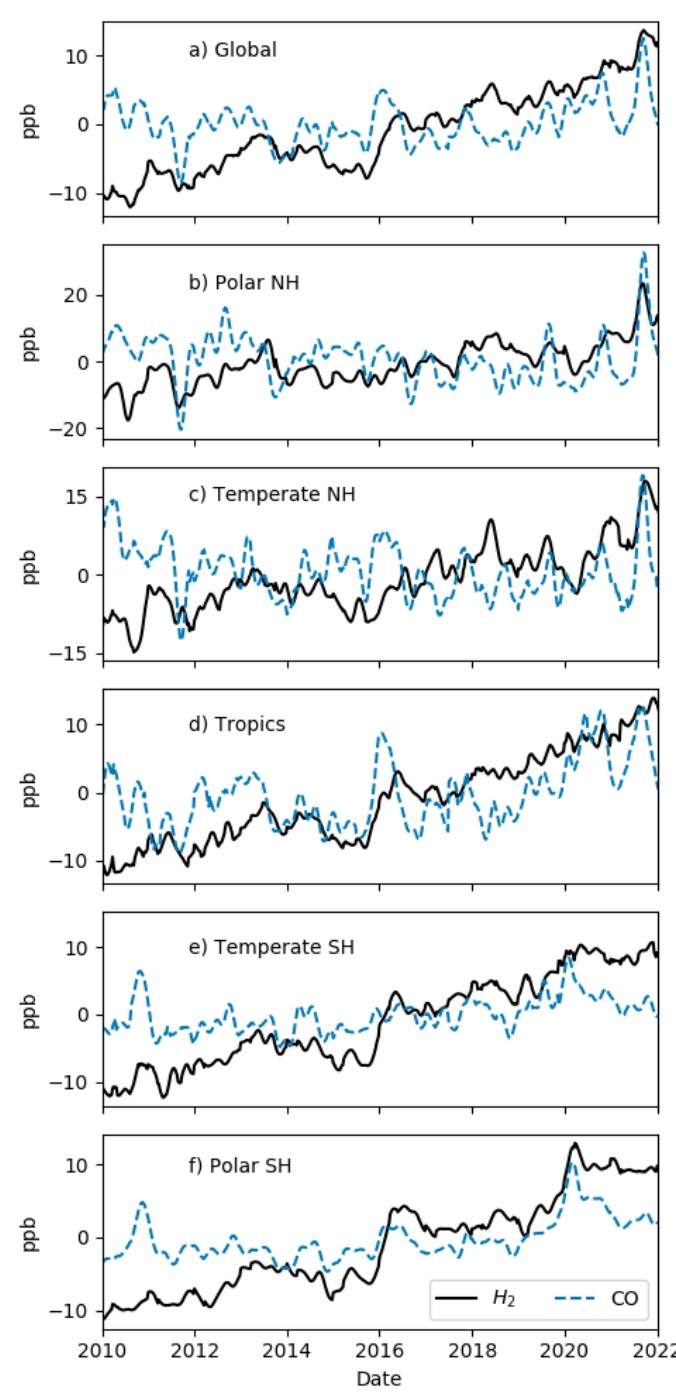
