# Peer review of "Atmospheric H2 observations from the NOAA Global"

_Atmospheric Measurement Techniques, 2024_

## Author Comment (AC1)

Responses to RC1: 'Comment on amt-2024-4', Simon O'Doherty, 28 Feb 2024

General Comments:

1. This is an important description detailing the calibration strategy used to be able to assign meaningful and traceable mole fraction values to a global network of $H_2$ flask measurements and is exactly the type of manuscript that should be published in AMT. The difficulty in this "warts and all" description of how the calibration procedures have developed over time is that it makes for quite difficult reading due to the complex nature of the many different tank comparisons performed using many different instruments. I can't recommend a better way of presenting the data, because ultimately all the useful information is contained within the manuscript and SI. The reader will just have to persevere, jumping between text, Figures, Tables and SI to find what is immensely useful information for setting up a calibration procedure for $H_2$

Thank you for your detailed review. Your comments and questions are very helpful.

We agree that the manuscript and the SI are covering a lot of information. The WMO $H_2$ calibration scale adoption and transfer was a long and iterative effort to make the most of existing measurements. We have moved 3 Tables to the Supplementary Information file and removed some redundant text in the main manuscript.

2. Section 3 of the manuscript describes the data quality assurance and quality control of the ~6000 glass flasks that have been collected at a global network of sampling sites between 2009-2021. This is an immensely impressive and useful dataset. I was a little surprise however, that this manuscript describing the analytical detail is being published after a paper whew the measurements have been used to assess the representation of the H2 atmospheric budget in the state-of-the-art GFDL-AM4.1 global atmospheric chemistry climate model (Paulot, F et al., 2023).

I apologize for the timing of this paper's submission. I very much underestimated the time it would take to get co-authors' comments back and then the manuscript had to go through a new internal review procedure before submission. The revised $H_2$ measurements from the Cooperative Global Surface Sampling Network were made available on the NOAA GML public ftp in May 2023. The work and paper led by Fabien Paulot moved fast and we did not want to delay the publication of the modeling findings as there is growing interest among policymakers to understand where the science stands.

3. It is clear from the calibration work that has been undertaken by NOAA, that aluminium cylinders are not stable for $H_2$. This was recognised by NOAA many years ago and is why the primary calibration standards are filled into stainless steel electropolished Essex cylinders. However, even with this knowledge this hugely

important global network for H2 measurements has persisted in using aluminium cylinders for secondary and tertiary analysis and then tried to correct for the many different rates of calibration tank drifts. The paper details extensive problems using this approach (under-sampled cylinders, massively different rates of drift on a tank-to-tank basis), all of which propagates uncertainly into the measurements. Why has a different style of tank not been used, which does not suffer from these issues? I realise that Essex tanks (or a similar style of stainless-steel tank) are expensive but surely it is a requirement of a global H2 network to reduce measurement and calibration uncertainties where practicable by using tanks that don't drift?

Aluminum cylinders work well for $CO_2$, $CH_4$, $N_2O$ and $SF_6$ which have held higher priority historically (and presently) at NOAA GML. Aluminum cylinders are also cheaper and easier to use than stainless steel electropolished Essex cylinders.

A while back, the decision was made to continue using those cylinders for $H_2$ (and CO) calibration standards and to track their drifts regularly enough to correct for them. Adopting the $H_2$ gravimetric mixtures prepared in electropolished stainless steel cylinders as our primary standards after they were calibrated by the MPI was a first step. We are still evaluating options to improve the robustness of the $H_2$ calibration scale transfer. The existence and value of the NOAA $H_2$ dataset for recent years are becoming more known and we are working on securing some funding to buy more Essex cylinders.

4. I am a little unsure of the purpose of section 3.2.3, the text does not really indicate why the SPO measurements are given their own section (unless the point is to state that H2 stores well in glass flasks and SPO is uninfluenced by emission sources)?

Thank you for this clarification question. Your assumptions for the reasons are correct. We have removed some extraneous information in this section and added this sentence to connect with the WMO comparability goal:

"The average of the absolute differences for $H_2$ in SPO flask paired samples is less than 2 ppb ($\sigma \le 1.3$ ppb) and methods S and P $H_2$ pair averages at SPO agree within 1 ppb on average ($\sigma \le 1.7$ ppb)."

Specific comments:

1. defined the calibration scale as a WMO scale, whilst L30 defines it a the MPI scale, this is a little confusing so early on in the paper.

   This has been corrected.

2. Grey H2 not Gray H2

   Sorry, we use the American English spelling.

3. Novelli et al. [1991, 1992] not [1992, 1991]

This has been corrected.

4. 10-200ppb not 10s

This has been corrected.

5. Why are the Essex cylinders filled with dry air. I understand that Essex cylinders tend to be stable for $H_2$, however, is there any evidence that drying the air is a requirement for $H_2$ stability? In my experience Essex tanks filled with undried ambient air are also stable.

The GML $H_2$ primary standards in Essex cylinders were prepared gravimetrically by Brad Hall. They were not prepared with ambient air. We do not have experience with $H_2$ in humidified Essex tanks. We began testing dry air in Essex cylinders after hearing from colleagues that Essex tanks may be stable when filled dry.

6. What is NOAA going to do with the pre-2009/10 ambient air record for $H_2$

At this point we are focusing on maintaining the quality of the NOAA $H_2$ measurements going forward.

Sadly, the pre 2009/2010 $H_2$ measurements cannot be revised for reasons detailed in Supplementary text S1. These measurements are marked as rejected in the NOAA GML database and future NOAA $H_2$ data releases will not include them.

7. L272-276. What caused the tail or noisy baseline? Do you think use of peak height might have caused a bias; what effect did the higher grade of helium have (removed the noise/tail)? Do you use peak height or peak area for the data using the higher-grade helium, are peak height and peak area data comparable now? Why did it take 4-years to decide to use cleaner helium?

The issue with the peak tail or noisy baseline was very (Airgas) He tank dependent and we found that peak height was less sensitive than peak area in those instances. Colleagues in GML were using Matheson Research Grade He for GC-MS systems and when we tested that He for $H_2$, the baseline and peak looked good. We get very similar results for peak height and peak area. We still use peak height.

8. The word "few" is not informative.

We have replaced "few" in this sentence below:

"GML has performed an H9 instrument response calibration followed by tank calibrations 2 to 3 a few times a year over a 10-14 day period each time." (Line 298 in final view file)

9. You state that typically H2 tertiary standards lasted less than a year. However. Figure 3a & b show that many of these tanks lasted much longer than 1-year and most drifted quite appreciably.

Figure 3a & b shows the calibration records for the MAGICC system tertiary standards. Three (out of 17) tertiary standards were used for more than 14 months (cf. Table 2):

- on H11: ND46735 had a small quadratic drift < 2.5 ppb/yr and was used for 17 months. ND38963 had a 6.2 ppb/yr linear drift and was used for 16 months.
- on H8: CA03409 had no detectable drift and it was used for 22 months.

We have revised the text in L 337-338 in final view file:

"Typically, the $H_2$ tertiary standards used during that time lasted less than a year and most displayed $H_2$ growth over time. "

to

"Out of 17 $H_2$ tertiary standards used during that time, 3 were used for more than 14 months and 14 displayed $H_2$ growth over time. "

10. Why only use 8 or the 11 standards

The three standards that are not used for the $H_2$ response curve of the MAGICC-3 system exhibit changing $H_2$ drift behaviors that are not captured well enough by their calibration records. For these 3 cylinders, the residuals of a best fit (quadratic) to their calibration histories span ranges beyond the range [-1.5 ppb, 1.5 ppb], in contrast with the other eight standards.

The suite of standards used for future $H_2$ response curves may change. Every 1 or 2 years, we will reevaluate the drift corrections and assignments for the 11 cylinders based on new calibration results. If residuals of a standard calibration history to a new best fit function are larger than 1.5 ppb or if $H_2$ grows beyond 700 ppb in a standard, we may decide to drop that cylinder from the suite of standards. We also may have to go beyond using a single linear or quadratic fit if the observed $H_2$ drift behavior for one standard will be better captured by a set of different functions.

One sentence was added in the main text (L 376-376 in final view file):

"The three cylinders that are not used exhibit H2 growth that is difficult to capture with periodic calibration episodes."

11. You now define the tanks as working tanks, not secondary or tertiary – why change the tank definition, it is confusing.

We use "working" standards to differentiate from true "tertiary" standards. Secondary standards are only used in GML to transfer the scale to tertiary standards. We are not using secondary standards for $H_2$ after April 2019.

Section 2 introduction states: "(...) we describe the GML tank air $H_2$ calibration system and the scale transfer from the primary standards to secondary and tertiary standards (2009-April 2019) or from the primary standards to working standards (after April 2019). The tertiary standards and working standards are used to calibrate the $H_2$ instrument response on the flask air analysis systems and value assign discrete air measurements."

12. Why change at 250 psia, is there evidence that the tank drifts at pressures below this?

A tank pressure of 250 psi is a cutoff GML uses as there are no or very small changes in the tank mole fraction for the GHG measured in GML, especially $CO_2$ [WMO. 2016; Schibig et al., 2018].

GML rarely uses standards with lower pressures. It did happen for example for CC305198 (A) and we noticed an acceleration in its $H_2$ drift rate (SI Table 2). The $H_2$ growth in aluminum tanks is suspected to be caused by a surface process (see next question's response and [Jordan and Steinberg, 2011]) and therefore the drift rate could be influenced by pressure in the tank.

Schibig, M. F., Kitzis, D., and Tans, P. P.: Experiments with CO2-in-air reference gases in high-pressure aluminum cylinders, Atmos. Meas. Tech., 11, 5565–5586, https://doi.org/10.5194/amt-11-5565-2018, 2018.

WMO: 18th WMO/IAEA Meeting of Experts on Carbon Dioxide Concentration and Related Tracers Measurement Techniques (GGMT-2015), La Jolla, CA, USA, 13–17 September 2015, GAW Report No. 229, World Meteorological Organization, Geneva, Switzerland, 2016.

13. Do you know why H2 drifts in air filled aluminium cylinders? If a non-drifting tank is reused, is it still non-drifting and vice versa with a drifting tank?

Jordan and Steinberg, AMT, 2011 (section 3.1, Figure 5) discuss the stability of reference air in various high pressure cylinder types. They analyzed > 100 cylinders over 1-6 years and found that "highly variable storage properties were observed in aluminium cylinders." They go into more details about cylinders made with different alloys and propose that different alloys and manufacturing processes may impact the integrity of the cylinder surface.

We do not have a lot of repeated fills for tanks used for $H_2$ work. For TST air cylinders which are refilled regularly and have 4 or more tank calibrations, it seems that AL47-104 and AL47-108 always exhibit significant drift in H2 for the 2 or 3 fills plotted, while AL47-113 shows no drift for 3 successive fills and AL47-145 had a very large drift for fill E and more

moderate drifts for fills F and G. We are paying attention to this issue and hope to understand more soon to avoid cylinders with large drifts.

14. If the tank shows signs of large initial growth in the first 0.5-2 years, why not fill then store a tank for this time before use?

Yes, it seems that waiting at least 2 years could help with some tanks. We now know we need to wait longer after a fill or document its behavior for a while before adopting them as a standard or a target tank. Our colleague MM has been screening cylinders over several months with regular analysis in the flask lab to pick reference air tanks with ambient level and stable $H_2$.

13. L451-452. I assume the three tanks are aluminium filled with dry air? – this information is not detailed in the text or Figures.

Yes this is correct, the MENI cylinders are 10 L Luxfer UK aluminium cylinders (AA6061) filled with dried air.

This information has been added, L 430.

15. SI L281. Figure 5 (a) is missing.

I am very sorry. The missing figure has been added. Thank you for noticing.

16. SI Figure 5. To understand the year in year comparisons it would be useful for the to have the error bars plotted.

See below, merged answer with the next question.

17. SI Figure 5. The data in 5(b) are not that easy to understand. Why are the NOAA (2018) and MPI (2019) data carried out a year apart quite similar, but the NOAA (2021) and MPI (2022) a year apart quite different (~2 ppb). There are also very few NOAA data points to compare with MPI.

The MPI MENI tanks go to other laboratories besides NOAA for analysis of a suite of gases ($CO_2$, $CH_4$, $N_2O$, $SF_6$, $CO$, $H_2$) and $CO_2$ stable isotopes ($^{13}C$, $^{18}O$). Delays can happen. We have added the reported reproducibility as error bars to the plots. The MPI BGC GC-RGA measurements (April 2020) have larger standard deviations and that instrument has a reported reproducibility of 2 ppb compared to the 0.5 ppb reported for the MPI BGC GC-PDD measurements.

18. L 462-463. You state the MENI program provides an important on-going check from MPI X2009 H2 calibration scale transfer in GML. What is not clear is how the results presented in SI Figure 5 are used?

As you point out these measurements are not very frequent but they still provide an independent on-going comparison directly with the CCL. If we ever see large differences, we will know we need to investigate and fix a problem. We replace the word "important" with "valuable" in the main article section 2.3.2.

19. Does the restating the information about the flask sampling systems need its own section (3.1), why can't the information be contained in Section 3.

We have eliminated section 3.1 to reduce redundancy.

20. Is there any indication that H2 is stable/not stable in the glass flasks over time?

GML has not performed long storage tests on flask samples in a while, and it is something we know we need to do.

To try to answer your questions, I am looking at air samples collected close in time but not analyzed at the same time. Note that the 2 flasks from a pair (collected at the same time at a site) are always analyzed back to back on the flask analysis system in GML.

The GML team in Hawaii often collects 2 flask pairs back to back at KUM. Here we look at 2 pairs collected within 1 hour of each other and results from the H11 instrument . The second pair is typically used for various types of testing. Below we look at the mean $H_2$ for each pair (top plots) and plot the pair mean differences as a function of the length of time between the analysis times of both pairs. The difference plot shows the mean $H_2$ for the pair analyzed later minus the mean $H_2$ for the pair analyzed earlier. The scatter likely reflects both short term variability in the ambient level at the site and uncertainty in the measurement.

[Figure]

[Figure]

This is a similar plot as above for H2 in the South Pole S and P flasks.

We only have 3 sampling dates for SPO with pairs analyzed more than 2 months apart.

In those 3 pairs, the flask pair mean difference is about 2 ppb, which means $H_2$ in the flasks analyzed later (2 P pairs and 1 S pair) are about 2 ppb higher than the flasks analyzed earlier.

So with the data on hand there is no evidence of a flask storage effect on $H_2$ as the scatter of the mean pair differences is comparable for different storage times.

21. L677 to 678 and Figure 7. How are reliable results between S and P methods defined and tested? Visually from Figure 7, it looks like the S flask data are slightly below the P flask data (looking at the apex of the annual cycles in 2020 and 2021 for example)?

Below is a plot of SPO $H_2$ S flask pair average minus P flask pair average at full resolution in green, with annual means of the differences shown in blue. Most of the time it looks noisy around zero and sometimes the differences are mostly positive. It is not clear what exactly causes these changes. We rotate the staff at the observatory and it may be due to slightly different sampling operations. The annual

mean difference ranges between -0.8 (2015) and 0.2 ppb and the standard deviation ranges between 0.6 (2021) and 1.7 (2010 and 2019) ppb.

[Figure]

22. How do you define "reliable", this is a bit non-specific.

This sentence has been removed.

23. What metrics have you used to determine that there are no biases?

The plot below shows $H_2$ at the times of transition between SPO flask shipments. The shorter storage samples (red symbols) were collected typically in late December or early January and were analyzed in the next 1.5 to 3 months. The longer storage samples (blue symbols) were collected in mid January to mid February and were analyzed 11 to 12 months later. We chose to limit the transition to 3 weeks of sampling dates, centered on the first sampling date with the longer storage time. The transition occurs when $H_2$ is increasing before peaking in February. There is no systematic offset for $H_2$ from the longer storage samples.

[Figure]

24. L779-781. It is clear that the Mauna Loa data show more short-term variability that Samoa and South Pole, but not necessarily Barrow

    Yes, this is correct.

25. L779 to 787. It is not clear how the maxima and minima for each site have been determined, and wouldn't these vary year to year given that there is a growth trend in the data?

    Thank you for your question. We used the smooth curve fit to the data and here we are talking about absolute min/max for each year so yes it includes the "trend". We have switched the order of sections 4.1 and 4.2 to introduce the curve fit and smooth curve concept before using it for the observatories extrema discussion. We have also switched the order for Figures 9 and 10.

26. What does ASC stand for?

    Sorry, this is the code for Ascension Island, and this has been clarified in the main text.

27. L 816 and Figure 10. Given all the sites are defined at the top of the plot, why do you need to use the x-axis to number the sites. Surely you should use it to illustrate the latitudinal gradient.

    The index labels on the x axis in this figure (Figure 9 now) have been removed.

    A sentence Line 728-729 mentions the interhemispheric gradient:

"The interhemispheric gradient of $H_2$, with higher levels in the SH, is apparent in the annual means

distribution across sites in Figure 9 (green circles)."

28. L 817 to 834. I can't find any information in the manuscript of SI defining the site acronyms or detailing their lat/longs (useful). Can this be contained in a Table? Also, in the text you define some sites described in the text e.g., TPI site, on Taiping Island, but don't define others e.g., TAP, AMY, LLN, CPT, KUM, WIS.

SI Table S4 has been added with this information in the Supplementary material. Thank you for the suggestion.

The paragraph about the sampling location change at KUM and WIS was removed.

We have added the country for the other sites in L 733.

"A few sites (for ex. TAP (Taiwan), AMY (Republic of Korea), LLN (Taiwan), CPT (South Africa)) show higher smooth curve annual maxima (Figure 9, red crosses), likely reflecting upwind local or regional emissions."

29. L828 to 834. Is this short description of moving sites required? There is no supporting evidence to explain why the mean level of H2 or seasonal cycle have changed since the move. Just the assumption that increased soil uptake is responsible – is the new location more inland? Can you use ozone deposition or radon measurements to confirm this?

We have removed this section discussing the change in sampling site locations.

We do not have colocated ozone or radon measurements at these sites to investigate the assumption of the soil sink impacting the measurements. Field studies are needed to advance the parameterization and estimation of the soil sink in various ecosystems and regions.

30. L844. A large proportion of Africa (and fires) are in the NH. The flask sites at ASC, NMB & CPT look well located to sample SH fires from the Peoples Republic of Congo, Angola and Zambia?

The plot below shows the $H_2$ and CO records at the 3 sites ASC, NMB and CPT. There are several samples at ASC and NMB in June-October months with elevated CO which may be related to biomass burning in South African countries. We do not see clear $H_2$ enhancements at the 3 sites during the region's fire season. Further analysis using atmospheric transport models and biomass burning products is

needed to study the observed variability.

[Figure]

31. Table 2. The time of use and Fill date time formats are different, less confusing if you use the same format.

The inconsistency has been fixed in Tables 1 and 2. Thank you for bringing it up.

---

## Author Comment (AC2)

Responses to RC2: 'Comment on amt-2024-4', Anonymous Referee #2, 06 Mar 2024  reply

This very thorough and detailed manuscript addresses the recalibration of the extensive 12-year NOAA global atmospheric hydrogen ($H_2$) measurement program to the MPI X2009 calibration scale.  It does this by adjusting for the significant and variable rates of growth of $H_2$ in the secondary and tertiary standard aluminum calibration tanks used for these measurements that were originally based on the NOAA X1996 calibration scale.  As the manuscript notes, this has become immensely important work as mankind moves increasingly toward $H_2$ as an energy storage medium that comes with a large potential to leak into the atmosphere and fundamentally alter the oxidation capacity of the atmosphere with respect to lengthening the atmospheric lifetimes the of methane and other anthropogenic greenhouse gases.

While acknowledging this importance, I had considerable difficulty reading the manuscript because of its distractingly minute detail.  One example is the inclusion of the serial numbers of the instruments used to make the measurements.  Another is the extensive comparisons of the performance of the three different "MAGICC" instrument systems used to make these measurements over time.  This is too much detail for the average reader's attention span, even though it is important that it be recorded somewhere.

One solution would be for the main paper to outline the principles of what was done with a few illustrative examples, and to move the bulk of the details that should be recorded somewhere either to the supplementary information (SI) addendum together with this AMT paper, or to a separate project report published by NOAA and available through the NOAA GML website.  I recommend distilling the main paper to something like one third to one half of its present length, while still conveying the rigor of this important work, including a few examples, and providing the full details elsewhere.

This paper aims to fully document the recent revisions affecting the NOAA 2009/2010-present $H_2$ measurement records, which together make the largest global data set for $H_2$. We have moved 3 tables to the SI and removed some duplicative or extraneous text.

Thank you for your advice to streamline some of the information or move some content to the SI to not burden the reader.

A few more specific comments follow:

1)  Is NOAA GML now using stable stainless steel gas cylinders to propagate its calibrations going forward so that drift adjustments resulting from the use of aluminum cylinders will no longer be an issue?

GML uses multi gas standards in aluminum cylinders in the flask analysis laboratory. We are still exploring options as explained in Reviewer 1, Question #3.

2)  The dry air mole fraction ppb $H_2$ concentration units are not defined when they are first used.

The definition of the mole fraction unit (ppb) has been added in the introduction.

3)  Please explain why calibration scales are necessary for atmospheric research, as compared to using individual calibration standards that are not related to a specific scale.  This important point is not widely appreciated, especially among national metrology institutes.

We have added the text below at the beginning of section 2 in the main manuscript.

> To reliably infer fluxes and changes in fluxes over time from atmospheric measurements, scientists need to detect small temporal and spatial gradients in the abundance of trace gases. This requires comparable data across time and across monitoring networks to ensure biases are minimized and do not influence interpretation. The use of a common calibration scale among measurement laboratories ensures data are traceable to a common reference. It is the first step in preventing biases that could stem from using different references.

4)  The word "best", used in line 170 to describe the post-2009 NOAA $H_2$ data, is subjective.  A better term might be "most precise", or something to that effect.

"Best" as been replaced in the sentence below:

> The  more precise and better calibrated NOAA $H_2$ measurement records date back to 2009/2010 and are the main focus of this paper.

5)  Following on the above comment, NOAA prepared their X1996 calibration scale well before the measurements that are recalibrated in this paper.  Are there pre-2009 NOAA $H_2$ data that could still benefit from being recalibrated to the MPI calibration scale despite their lower precisions?

Unfortunately no. Please see SI section S1 and SI Figures 12 & 13  for our detailed explanation.

6)  Should Paul Novelli, and perhaps Ed Dlugokencky, be coauthors of this work?  They are both retired from NOAA, but much of this work was done by them.

Paul Novelli retired in 2017 and he was the science lead that started the CO and H2 measurements at NOAA. The paper references Paul's early work and papers and it makes it clear the NOAA $H_2$ work originated with him. Co-author Andy Crotwell streamlined the calibration scale propagation and Andy Crowell and Brad Hall worked with Paul on the adoption of the new instrumentation and the preparation of the new gravimetric mixtures in Essex tanks (Novelli et al., 2009). None of this work would have been possible without Andy's analytical and technical expertise and leadership. Andy implemented a strict calibration standards hierarchy for CO and $H_2$ and he led the adoption of best practices to store and manage data files. Andy has been the technical lead in charge of the calibration work for multiple gases ($CO_2$, $CH_4$, CO, $H_2$) at NOAA GML for many years, working closely with Brad Hall and others. Ed Dlugokencky

retired from NOAA GML in spring 2023. He was not involved with the $H_2$ measurements or the CCL calibration scale adoption. We reference two papers by Ed when we give background information about the NOAA GHG measurements in samples from the Cooperative Global Air Sampling Network.

The COVID-19 pandemic restricted lab access for many months and provided Andy and several of us in GML time to work on the $H_2$ dataset. It was a group effort and the NOAA GML and CIRES staff involved as well as our close collaborators on $H_2$ in other laboratories (MPI BGC, CSIRO, UCI) are co-authors on the paper.

7) The Teflon o-rings used in the flask stopcocks (line 568) are highly permeable to $H_2$ and other gases. Since the flasks are pressurized and $H_2$ permeates much faster than $N_2$ and $O_2$, shouldn't the $H_2$ mole fraction decrease with time between sampling and analysis? Has this been tested?

The global network flasks are pressurized to 17-19 psia. We have not conducted systematic storage tests to evaluate if the permeability of teflon affects the integrity of $H_2$ in the air samples. The GML measurement team plans to do sampling equipment and flask storage tests in the not too distant future to carefully evaluate the stability of a suite of gases.

For now, we have looked at samples collected close in time, and analyzed at different times leading to different storage times. Please see the response to Reviewer 1, Question 20.

8) Please use the widely accepted spelling of "gases" instead of "gasses". Please also change "data is" to "data are" since "data" is the Latin plural of "datum".

Changes have been made. Thank you for pointing this out.

9) Given the range of coauthors and institutions involved in this work, and the number of years of data that are corrected, I assume that the acknowledgments and the financial support (lines 958-966) may be incomplete.

Please see the earlier response to your question 6.

Summary:

This is important work that should be published, and AMT is an appropriate venue for this. But it should first be distilled to a more readable structure that conveys the principles of what was done and summarizes the results, with the details presented either in the SIs or in a separate reference publication.